# The glycosomal ATP-dependent phosphofructokinase of *Trypanosoma brucei* operates also in the gluconeogenic direction

Nicolas Plazolles[1], Hanna Kulyk[2,3], Edern Cahoreau[2,3], Marc Biran[4], Marion Wargnies[1], Erika Pineda[1], Mohammad El Kadri[1], Aline Rimoldi[1], Perrine Hervé[1], Corinne Asencio[1], Loïc Rivière[1], Paul A. M. Michels[5], Daniel Inaoka[6,7,8], Emmanuel Tetaud[1], Jean-Charles Portais[2,3,9], Frédéric Bringaud [1]*

**1** Univ. Bordeaux, CNRS, Microbiologie Fondamentale et Pathogénicité (MFP), UMR 5234, Bordeaux, France, **2** Toulouse Biotechnology Institute, Université de Toulouse, CNRS, INRA, INSA, Toulouse, France, **3** MetaToul–MetaboHUB, Toulouse, France, **4** Univ. Bordeaux, CNRS, Centre de Résonance Magnétique des Systèmes Biologiques (CRMSB), UMR 5536, Bordeaux, France, **5** School of Biological Sciences, The University of Edinburgh, Edinburgh, Scotland, **6** Department of Molecular Infection Dynamics, Institute of Tropical Medicine (NEKKEN), Nagasaki University, Nagasaki, Japan, **7** School of Tropical Medicine and Global Health, Nagasaki University, Nagasaki, Japan, **8** Department of Biomedical Chemistry, Graduate School of Medicine, The University of Tokyo, Tokyo, Japan, **9** STROMALab, Université de Toulouse, INSERM U1031, EFS, INP-ENVT, UPS, Toulouse, France

\* frederic.bringaud@u-bordeaux.fr

## Abstract

In the glucose-free environment of the midgut of the tsetse fly vector, the procyclic forms of *Trypanosoma brucei* primarily consume proline to feed its central carbon and energy metabolism. In this context, the parasite produces through gluconeogenesis, glucose 6-phosphate (G6P), the precursor of essential metabolic pathways, from proline catabolism. We show here that the parasite uses three different enzymes to perform the key gluconeogenic reaction producing fructose 6-phosphate (F6P) from fructose 1,6-bisphosphate, (*i*) fructose-1,6-bisphosphatase (FBPase), the canonical enzyme performing this reaction, (*ii*) sedoheptulose-1,7-bisphosphatase (SBPase), and (*iii*) more surprisingly ATP-dependent phosphofructokinase (PFK), an enzyme considered to irreversibly catalyze the opposite reaction involved in glycolysis. These three enzymes, as well as six other glycolytic/gluconeogenic enzymes, are located in peroxisome-related organelles, named glycosomes. Incorporation of $^{13}$C-enriched glycerol (a more effective alternative to proline for monitoring gluconeogenic activity) into F6P and G6P was more affected in the PFK null mutant than in the FBPase null mutant, suggesting the PFK contributes at least as much as FBPase to gluconeogenesis. We also showed that glucose deprivation did not affect the quantities of PFK substrates and products, whereas an approximately 500-fold increase in the substrate/product ratio was expected for PFK to carry out the gluconeogenic reaction. In conclusion, we show for the first time that ATP-dependent PFK can function *in vivo* in the gluconeogenic direction, even in the presence of FBPase activity. This particular

**Data availability statement:** All the numerical data, as well as the original, uncropped, and minimally adjusted images are available from the Zenodo database (https://doi.org/10.5281/zenodo.15148560).

**Funding:** FB was supported by the Centre National de la Recherche Scientifique (CNRS, https://www.cnrs.fr/), the Université de Bordeaux (https://www.u-bordeaux.fr/), the Agence National de Recherche (ANR, https://anr.fr/) ADIPOTRYP (grant number ANR-19-CE15-0004-01), and TRYPADIFF (grant number ANR-23-CE15-0040-01), the Laboratoire d'Excellence (https://www.enseignementsup-recherche.gouv.fr/cid51355/laboratoires-d-excellence.html) through the LabEx ParaFrap (grant number ANR-11-LABX-0024), and the "Fondation pour le Recherche Médicale" (FRM) ("Equipe FRM", grant n°EQU201903007845). The funders had no role in study design, data collection and analysis, decision to publish, or preparation of the manuscript.

**Competing interests:** The authors have declared that no competing interests exist.

**Abbreviations :** BIC, Bordeaux Imaging Center; BSD, blasticidin; BSF, bloodstream forms; CFP, cyan-fluorescent protein; DHAP, dihydroxyacetone phosphate; E4P, erythrose 4-phosphate; FBPase, fructose-1,6-bisphosphatase; FHc, cytosolic fumarase; FLIM, fluorescence lifetime imaging microscopy; FRDg, glycosomal NADH-dependent fumarate reductase; FRET, fluorescence resonance energy transfer; F1,6BP, fructose 1,6-bisphosphate; F6P, fructose 6-phosphate; GAPDH, glyceraldehyde-3-phosphate dehydrogenase; GFP, green fluorescent protein; GK, glycerol kinase; Gly3P, glycerol 3-phosphate; GPDH, NADH-dependent glycerol-3-phosphate dehydrogenase; G3P, glyceraldehyde 3-phosphate; G6P, glucose 6-phosphate; G6PDH, glucose-6-phosphate dehydrogenase; HK, hexokinase; H6P, hexose 6-phosphate; $^1$H-NMR, proton nuclear magnetic resonance; IDT, Integrated DNA Technologies; MDH, glycosomal malate dehydrogenase; MID, Mass Isotopomer Distribution; M6P, mannose 6-phosphate; OAA, oxaloacetate; PAC, puromycin; PCF, procyclic forms; PEP, phosphoenolpyruvate; PEPCK, phosphoenolpyruvate

feature, which precludes loss of ATP through a futile cycle involving PFK and FBPase working simultaneously in the glycolytic and gluconeogenic directions, respectively, is possibly due to the supramolecular organization of the metabolic pathway within glycosomes to overcome thermodynamic barriers through metabolic channeling.

---

## Introduction

In microorganisms, glycolysis and gluconeogenesis are critical metabolic pathways that enable cells to adapt to fluctuating environmental conditions. Glycolysis is the primary pathway through which glucose is broken down to feed central carbon metabolism. Phosphofructokinase (PFK) plays a pivotal role in glycolysis by catalyzing the conversion of fructose 6-phosphate (F6P) to fructose 1,6-bisphosphate (F1,6BP) [1]. Conversely, gluconeogenesis produces glucose 6-phosphate (G6P), a key precursor feeding essential metabolic pathways, from non-carbohydrate precursors when external glucose is scarce. Fructose-1,6-bisphosphatase (FBPase, EC 3.1.3.11) is the key enzyme in gluconeogenesis in most organisms, irreversibly catalyzing the conversion of F1,6BP to F6P, effectively reversing the reaction catalyzed by PFK in glycolysis, but with the production of inorganic phosphate (Pi) instead of ATP [2]. Usually, the activities of PFK and FBPase are tightly regulated allosterically by multiple metabolites to avoid the futile cycle where both pathways operate simultaneously, with a net loss of one ATP molecule at each cycle [3]. This regulation is crucial for microorganisms, especially in environments where nutrient availability is subject to fluctuations, allowing them to efficiently manage their energy resources and maintain metabolic flexibility [4].

Two main evolutionary groups of PFK are distinguished by their phospho-donor substrates. The ubiquitous ATP-dependent PFK (EC 2.7.1.11) studied here, uses ATP as the phospho-donor and is found in many bacteria and protists, plants, and all vertebrates [1]. The negative $\Delta G^O$ calculated from the kinetic parameters of the enzyme isolated from numerous organisms sustained the dogma that ATP-dependent PFK only acts in the glycolytic direction *in cellulo* [5]. Indeed, under steady-state conditions with cells grown in high glucose media, PFK functions as an almost exclusively forward-driven glycolytic step in all cells. For instance, the reverse gluconeogenic flux of PFK in kidney cells was estimated to be less than 0.7% of the glycolytic forward flux [6]. Whereas ATP-dependent PFKs catalyze a reaction that is essentially irreversible under physiological conditions, the reaction catalyzed by the second group of PFKs (inorganic pyrophosphate (PPi)-dependent PFK, EC 2.7.1.90) is reversible, could be close to equilibrium *in vivo* and is generally not regulated [7]. The PPi-dependent PFKs uses PPi as the phospho-donor, are only found in plants, certain protists, and certain prokaryotes (both Bacteria and Archaea), and can functionally replace both ATP-dependent PFK and FBPase [8]. Here, we address the role of the ATP-dependent PFK expressed in *Trypanosoma brucei* grown in glucose-depleted conditions.

*T. brucei* is an extracellular parasite that causes sleeping sickness in humans, a neglected tropical disease in sub-Saharan Africa, and nagana in animals. This parasite undergoes a complex life cycle from the bloodstream and tissues of a

kinase; PFK, phosphofructokinase; PGI, glucose-6-phosphate isomerase; PGLS, 6-phosphogluconolactonase; Pi, inorganic phosphate; PMI, phosphomannose isomerase; PPDK, pyruvate phosphate dikinase; PPi, inorganic pyrophosphate; PPP, pentose-phosphate pathway; PYK, pyruvate kinase; P5P, pentose 5-phosphate; RFP, red fluorescent mApple protein; Rib5P, ribulose 5-phosphate; RPI, ribose-5-phosphate isomerase; RuPE, ribulose-5-phosphate epimerase; R5P, ribose 5-phosphate; SBPase, sedoheptulose-1,7-bisphosphatase; S1,7BP, sedoheptulose 1,7-bisphosphate; S7P, sedoheptulose 7-phosphate; TAL, transaldolase; TIM, triose-phosphate isomerase; TKT, transketolase; X5P, xylose 5-phosphate; YFP, yellow-fluorescent protein; 1,3BPG, 1,3-bisphosphoglycerate; 2PG, 2-phosphoglycerate; 3PG, 3-phosphoglycerate; 6PG, 6-phosphogluconate; 6PGDH, 6-phosphogluconate dehydrogenase; 6PGL, 6-phosphogluconolactone.

mammalian host (bloodstream forms [BSF]) to the alimentary tract (procyclic forms [PCF]) and the salivary glands (metacyclic forms) of its blood-feeding insect vector (the tsetse). The complexity of *T. brucei*'s life cycle leads to the capacity for fast and high adaptation to environmental conditions, mostly through metabolic changes related to energy metabolism. One of the factors probably playing a role in these efficient changes is the presence of peroxisome-related organelles, called glycosomes, which contain the first six or seven glycolytic/gluconeogenic steps commonly present in the cytosol of other eukaryotic cells [9]. The unusual sequestration of glycolysis/gluconeogenesis in a membrane-bound compartment confers unique metabolic properties to the parasite. The most striking feature is the loss of much kinetic regulation of the main glycolytic kinases (PFK and hexokinase [HK], EC 2.7.1.1) as a result of the disconnection between the glycosomal and cytosolic ATP pools. Indeed, the glycosomal membrane is impermeable to large molecules such as nucleotides and bulky cofactors, which cannot be exchanged with the cytosol through the glycosomal pores which, by analogy to pores in other peroxisomes, probably allow free permeation of small solutes with Mr 300–400 Da [10,11]. Consequently, it is assumed that each molecule of ATP consumed within the glycosomes must be regenerated in the organelle. This is, for instance, the role of the glycosomal production of succinate and glycerol from glucose in PCF trypanosomes [12].

When grown *in vitro* in standard media, PCF *T. brucei* preferentially use glucose as a carbon source to fuel their central metabolism. Instead, in the glucose-depleted and amino acid-rich midgut environment of its insect vector, the parasites use proline as the main carbon source [13]. The parasite's rapid transition to glucose deprivation [14] suggests an efficient control of glycolytic/gluconeogenic enzymes, such as PFK and FBPase. As mentioned above, *T. brucei* ATP-dependent PFK, which evolutionarily originated from a PPi-dependent enzyme [15,16], has lost most kinetic regulation, with AMP remaining a moderate activator of the activity [17]. *T. brucei* also expresses a conventional FBPase and a sedoheptulose-1,7-bisphosphatase (SBPase, EC 3.1.3.37). It has been proposed that the latter is part of a cycle in the non-oxidative branch of the pentose-phosphate pathway (PPP), which produces F6P from glyceraldehyde 3-phosphate (G3P) and dihydroxyacetone phosphate (DHAP), and could therefore bypass FBPase activity [18] (Fig 1A). In addition, SBPase is structurally related to FBPase and may accept both F1,6BP and sedoheptulose 1,7-bisphosphate (S1,7BP) as substrates, as reported in other microorganisms [19–22]. PFK, FBPase, and SBPase are all located within the glycosomes [9,23] and are constitutively expressed in PCF and BSF (Fig 1B). It is noteworthy that FBPase activity is at the limit of detectability in total extracts of PCF and BSF [23–25].

Here, we have studied the role of FBPase, SBPase, and PFK in the gluconeogenesis of the PCF trypanosomes and showed that the ATP-dependent PFK contributed at least as much as FBPase to the production of G6P in glucose-depleted conditions. We have also shown that the concentration of PFK substrates and products does not favor its production of F6P from F1,6BP, suggesting that the glycosomal metabolism pathway is channeled to facilitate the gluconeogenic role of PFK.

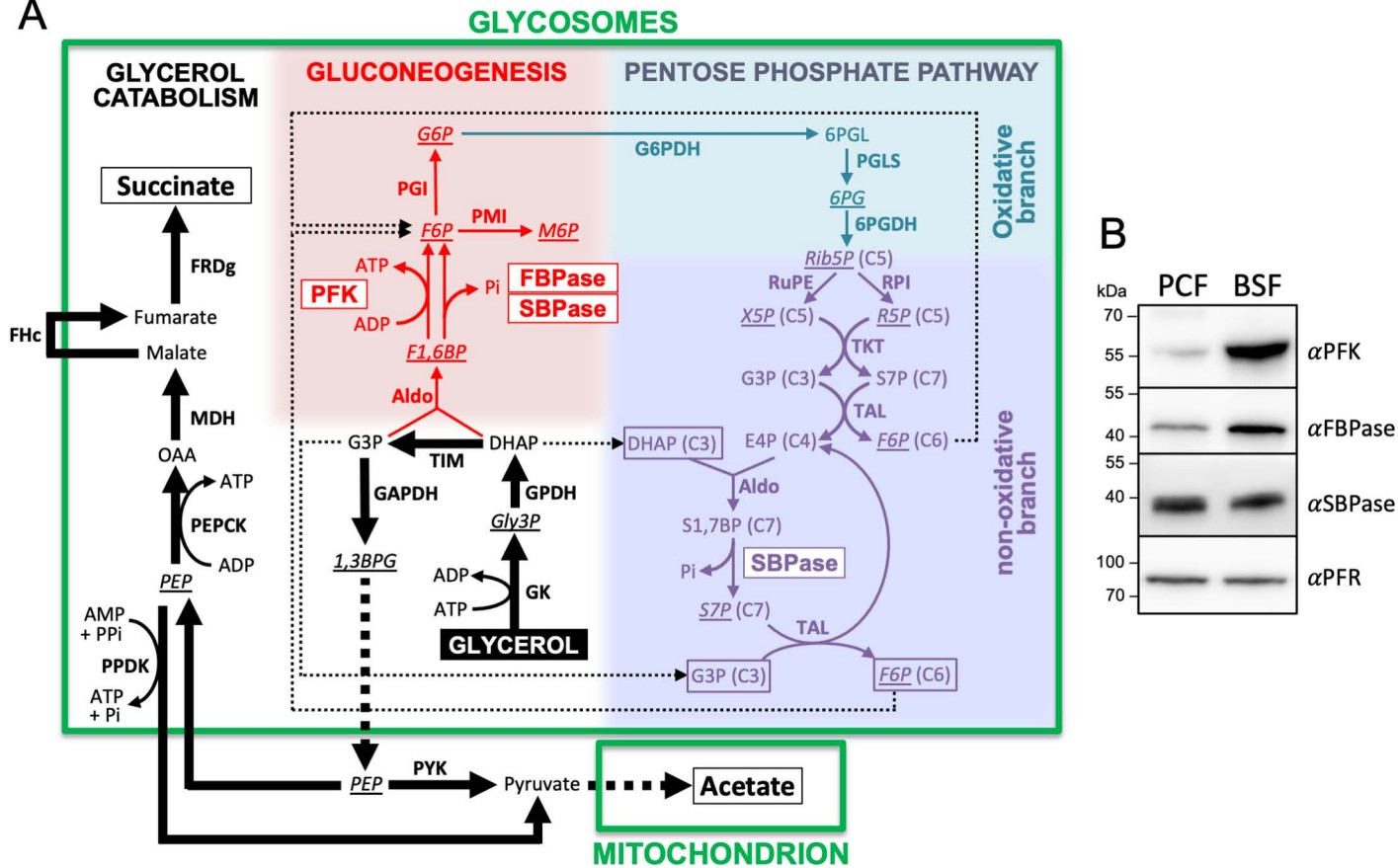

**Fig 1. Schematic representation of the glycerol metabolism of PCF trypanosomes.** This figure depicts glycerol metabolism rather than proline metabolism, since most metabolic analyses presented in the manuscript are performed with glycerol as the sole carbon source. Panel **A** shows glycerol metabolism leading to the excreted end products, succinate and acetate in the glycosomes and the mitochondrion, respectively, while gluconeogenesis (production for G6P from G3P and DHAP) is represented in red. The oxidative and non-oxidative branches of the pentose-phosphate pathway (PPP) are highlighted in blue and purple, respectively. The three addressed enzymes (PFK, FBPase, and SBPase), possibly responsible for production of F6P from F1,6BP, are boxed. It is noteworthy that the possible cycle in the non-oxidative branch of the PPP involving SBPase, which is supplied with DHAP and G3P (boxed) to produce F6P (boxed) [18], has not yet been demonstrated experimentally in trypanosomes. Metabolites analyzed by IC–MS/MS are underlined and in italic. 2PG and 3PG produced from 1,3BPG and precursor of PEP (dashed line) are not shown. For the sake of clarity, reversible reactions are not illustrated and neither the Gly3P/DHAP shuttle nor nucleotides and other cofactors are shown. The thin dotted lines highlight possible redistribution of G3P, DHAP, and F6P between gluconeogenesis and the PPP. Abbreviations: DHAP, dihydroxyacetone phosphate; E4P, erythrose 4-phosphate; F1,6BP, fructose 1,6-bisphosphate; 1,3BPG, 1,3-bisphosphoglycerate; F6P, fructose 6-phosphate; G3P, glyceraldehyde 3-phosphate; G6P, glucose 6-phosphate; Gly3P, glycerol 3-phosphate; 6PG, 6-phosphogluconate; 6PGL, 6-phosphogluconolactone; M6P, mannose 6-phosphate; OAA, oxaloacetate; PEP, phosphoenolpyruvate; 2PG, 2-phosphoglycerate; 3PG, 3-phosphoglycerate; Rib5P, ribulose 5-phosphate; R5P, ribose 5-phosphate; S7P, sedoheptulose 7-phosphate; S1,7BP, sedoheptulose 1,7-bisphosphate; X5P, xylose 5-phosphate. Indicated enzymes are: Aldo, aldolase; **FBPase**, fructose-1,6-bisphosphatase; FRDg, glycosomal NADH-dependent fumarate reductase; FHc, cytosolic fumarase; G6PDH, glucose-6-phosphate dehydrogenase; GAPDH, glyceraldehyde-3-phosphate dehydrogenase; GK, glycerol kinase; dehydrogenase; GPDH, NADH-dependent glycerol-3-phosphate dehydrogenase; MDH glycosomal malate dehydrogenase; PEPCK, phosphoenolpyruvate carboxykinase; **PFK**, phosphofructokinase; 6PGDH, 6-phosphogluconate dehydrogenase; PGI, glucose-6-phosphate isomerase; PMI, phosphomannose isomerase; PPDK, pyruvate phosphate dikinase; PYK, pyruvate kinase; RPI, ribose-5-phosphate isomerase; RuPE, ribulose-5-phosphate epimerase; **SBPase**, sedoheptulose-1,7-bisphosphatase; TAL, transaldolase; TKT, transketolase; TIM, triose-phosphate isomerase. Panel **B** shows a comparative analysis of PFK, FBPase and SBPase expression in PCF and BSF trypanosomes by western blot, using paraflagellar rod (PFR) expression as loading control. The data underlying this figure can be found in https//doi.org/10.5281/zenodo.15148560.

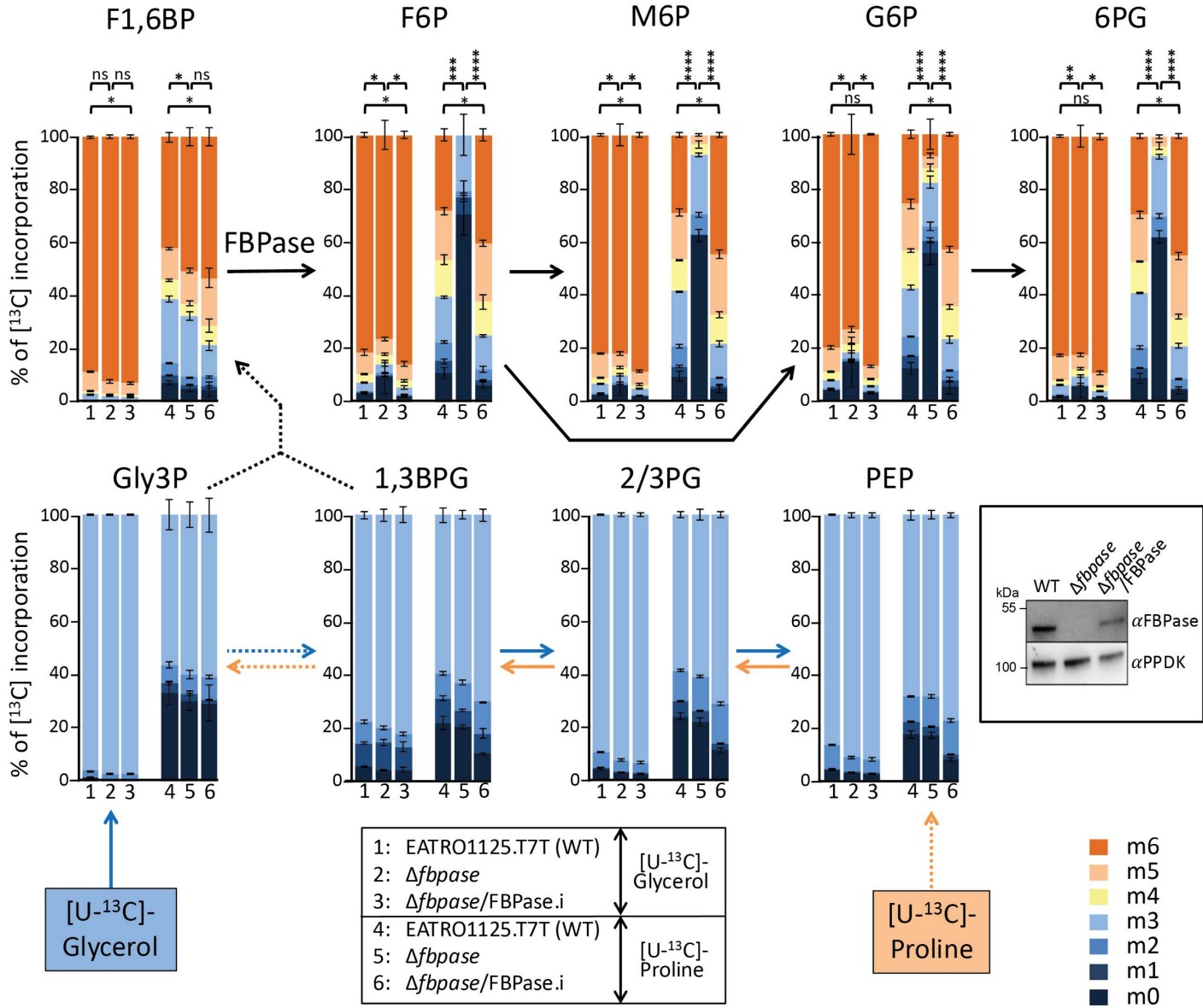

**Fig 2. Comparative IC–MS/MS analysis of intracellular metabolites after isotopic labeling with [U-13C]-proline or [U-13C]-glycerol.** The EATRO1125.T7T parental (WT), Δ*fbpase* and tetracycline-induced Δ*fbpase*/FBPase (Δ*fbpase*/FBPase.i) cell lines were incubated for 2 h in PBS containing 2 mM [U-13C]-glycerol (columns 1–3) or 2 mM [U-13C]-proline (columns 4-6) prior to metabolite extraction. The figure shows enrichment of key gluconeogenic intermediates at 0 to 6 carbon positions (m0 to m6, with a color code on the side) with [13C]-incorporation expressed as percentage of all corresponding molecules (MID; Mass Isotopomer Distribution). The color of the arrows corresponds to metabolism of glycerol (blue), proline (orange) or the two 13C-enriched carbon sources (black). The solid and dotted arrows correspond to a single or at least two enzymatic steps. For abbreviations, see Fig 1. 2/3PG means that 2-phosphoglycerate (2PG) and 3-phosphoglycerate (3PG) are undistinguished by IC–MS/MS. The inset shows a western blot analysis of the three cell lines with the anti-FBPase and anti-PPDK (control) immune sera. Data obtained with [U-13C]-proline, previously presented in Wargnies and colleagues [23], have been included here for comparison with the labeling with [U-13C]-glycerol. Each value corresponds to the mean of three independent experiments (biological replicates) and a *t* test was performed by assuming equal variances to determine the *p*-value (*p < 0.1; **p < 0.01; ***p < 0.001; ****p < 0.0001; ns, not significant). The data underlying this figure can be found in https//doi.org/10.5281/zenodo.15148560.

## Results

### Optimization of gluconeogenesis analysis

We previously showed that the production of hexose 6-phosphates (H6P) from proline by PCF trypanosomes is strongly impaired by the absence of FBPase (Tb927.9.8720) in the Δ*fbpase* mutant cell line and rescued in the Δ*fbpase*/FBPase.i cell line (.i standing for "tetracycline-induced"), which re-expresses an ectopic copy of the *FBPase* gene in the Δ*fbpase* background using the tetracycline-inducible pLew100 vector [23]. These data, previously described in [23] and included in Fig 2 (columns 4–6, labeling with [U-$^{13}$C]-proline), demonstrated that FBPase is involved in gluconeogenesis. Briefly, this was determined by ionic-exchange chromatography coupled with tandem mass spectrometry (IC–MS/MS) analyses of the incorporation levels of $^{13}$C atoms from uniformly [$^{13}$C]-enriched proline ([U-$^{13}$C]-proline) into key gluconeogenic intermediates. Incorporation of [$^{13}$C]-enriched atoms from proline into triose phosphates (PEP, 2/3PG, 1,3BPG, and Gly3P, see Fig 1 for abbreviations) and F1,6BP are equivalent in the parental, Δ*fbpase*, and Δ*fbpase*/FBPase.i cell lines. However, incorporation of [$^{13}$C]-enriched atoms in F6P (the product of the reaction catalyzed by FBPase), as well as in metabolites produced from F6P, *i.e.,* G6P, mannose 6-phosphate (M6P) and 6-phosphogluconate (6PG), was strongly reduced in the Δ*fbpase* mutant compared to the parental and rescue cell line, but was not abolished. This suggests the existence of an alternative step to FBPase producing F6P from F1,6BP. However, the residual levels of [$^{13}$C]-incorporation into H6P in the Δ*fbpase* cell line were too low to investigate these alternatives. To increase the efficiency of [$^{13}$C]-incorporation through gluconeogenesis, [U-$^{13}$C]-proline was replaced by [U-$^{13}$C]-glycerol in the same experimental set-up (Fig 2). Indeed, for the same incubation time in the presence of the $^{13}$C-carbon source (2h), [U-$^{13}$C]-glycerol is directly incorporated into gluconeogenic intermediates (see Fig 1), whereas incorporation of [U-$^{13}$C]-proline into gluconeogenic intermediates may be significantly delayed by the 9-step metabolic pathway, including 7 mitochondrial steps, leading to PEP from proline [14]. As expected, 91.0 ± 2.2% of F1,6BP in the parental cells was fully [$^{13}$C]-enriched (m6) from [U-$^{13}$C]-glycerol, compared to 48.5±5.9% from [U-$^{13}$C]-proline (Fig 2; F1,6BP; columns 1 *versus* 4) and incorporation of [U-$^{13}$C]-glycerol in F6P is significantly reduced in the Δ*fbpase* cell line (Fig 2; F6P; columns 1 *versus* 2). More importantly, the amounts of fully [$^{13}$C]-enriched H6P (m6) extracted from the Δ*fbpase* cell line were 40-times higher with [U-$^{13}$C]-glycerol than with [U-$^{13}$C]-proline (78.3±4.5% *versus* 2.0±4.0%) (Fig 2). This clearly confirmed that gluconeogenesis occurs in the absence of FBPase and that glycerol is much more efficient to feed gluconeogenesis than proline.

### PFK is an important contributor to gluconeogenesis

As ATP-dependent PFK from *T. brucei* (Tb927.3.3270), as well as human, have been recently shown to operate in the gluconeogenic direction *in vitro* [26], PFK expression was down-regulated by RNAi in the parental and Δ*fbpase* backgrounds. Western blot analyses showed that the tetracycline-dependent conditional expression was leaky since PFK expression was 95% and 98% down-regulated in the absence of tetracycline in the $^{RNAi}$PFK.ni and Δ*fbpase*/$^{RNAi}$PFK.ni cell lines, respectively, and after 3 days of induction, PFK expression was at most 1% of that of the parental cell line (Fig 3A). Unlike to the parental, $^{RNAi}$PFK.ni and Δ*fbpase*/$^{RNAi}$PFK.ni cells, tetracycline-induced $^{RNAi}$PFK.i and Δ*fbpase*/$^{RNAi}$PFK.i cells failed to grow and eventually died in glucose-rich medium (Fig 3B).

To determine the extent of impaired PFK activity in the $^{RNAi}$PFK and Δ*fbpase*/$^{RNAi}$PFK cell lines, we monitored glycolysis by quantifying the amounts of excreted [$^{13}$C]-enriched end products from [U-$^{13}$C]-glucose metabolism, which are mainly acetate and succinate, using proton nuclear magnetic resonance ($^1$H-NMR) spectroscopy [28]. The use of [$^{13}$C]-enriched carbon source allows us to distinguish between the products excreted from metabolism of [U-$^{13}$C]-glucose (doublet resonances labeled by red arrows in Fig 3C) and an uncharacterized internal carbon source (single resonances labeled by blue arrows in Fig 3C) [27]. As expected, the amounts of end products excreted from glycolysis are greatly reduced in the $^{RNAi}$PFK.ni and Δ*fbpase*/$^{RNAi}$PFK.ni cell lines, due to the RNAi leakage (Fig 3C). After tetracycline induction, the glycolytic activity is reduced by ~95% in the mutant cell lines, confirming that PFK activity is strongly affected but not abolished.

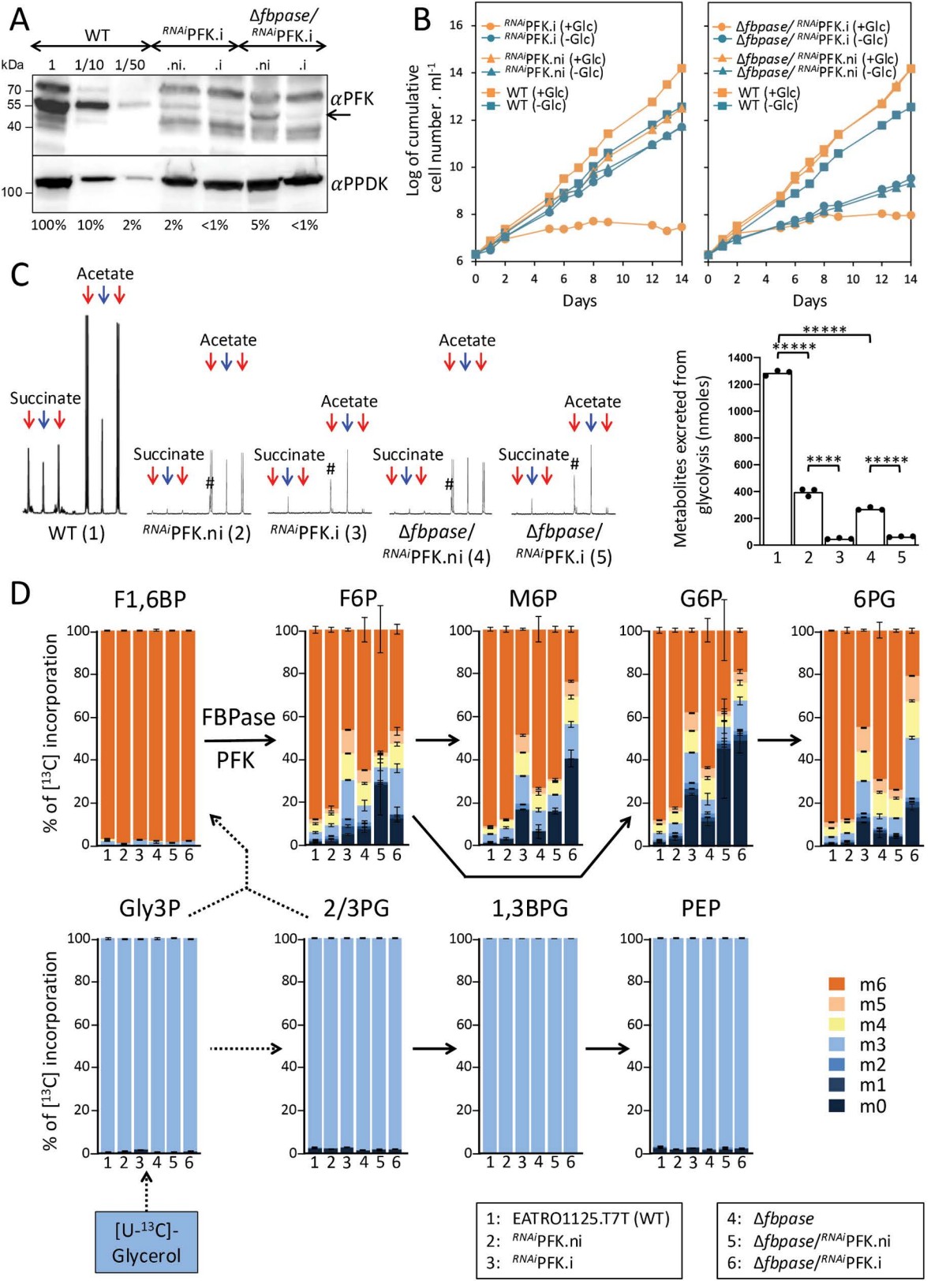

**Fig 3. IC–MS/MS analysis of intracellular metabolites extracted from FBPase and/or PFK mutant cell lines.** Panel **A** shows a western blot analysis of the EATRO1125.T7T parental (WT), Δ*fbpase*, as well as the tetracycline-induced (.i) and non-induced (.ni) *RNAi*PFK (clone A4) and Δ*fbpase*/*R-NAi*PFK (clone A9) cell lines using the anti-PFK and anti-PPDK (loading control) immune sera. Extracts of $5 \times 10^6$ cells have been loaded in all lanes, except lanes 1/10 and 1/50 which correspond to $5 \times 10^5$ and $10^5$ cells, respectively. The band corresponding to PFK is highlighted by an arrow and below the figure are the estimated amounts of PFK remaining, relative to WT cells (lane 1 with $5 \times 10^6$ cells) based on signal intensity measurements. Panel **B** shows growth curves of the PCF parental and mutant cell lines, tetracycline-induced or not, in glucose-rich (+Glc) and glucose-depleted (−Glc) conditions. Cells were maintained in the exponential growth phase (between $10^6$ and $10^7$ cells ml$^{-1}$), and cumulative cell numbers reflect normalization for dilution during cultivation. Similar growth curves were obtained for the *RNAi*PFK clone A8 and Δ*fbpase*/*RNAi*PFK clone F8 (S1 Fig). Panel **C** shows $^1$H-NMR analyses of succinate and acetate excreted after metabolism of [U-$^{13}$C]-glucose by the parental (WT) and mutant cell lines incubated in PBS containing 4 mM [U-$^{13}$C]-glucose. A part of each spectrum ranging from 1.6 to 2.6 ppm is shown in the left part. The doublet resonances corresponding to [$^{13}$C]-enriched molecules produced from [U-$^{13}$C]-glucose and the single resonances corresponding to non-enriched molecules produced from an unchar-acterized internal carbon source [27] are highlighted by red and blue arrows, respectively. The nature of the single resonance highlighted by an asterisk is unknown. Each spectrum corresponds to one representative experiment from a set of least three whose quantitative data are presented in graph on the right. Each value corresponds to the mean of three independent experiments (biological replicates) and a *t* test was performed by assuming equal variances to determine the *p*-value (*****$p < 0.00001$). In panel **D**, the parasites were incubated for 2 h in PBS containing 2 mM [U-$^{13}$C]-glycerol prior to metabolite extraction. The figure shows enrichment of key gluconeogenic intermediates at 0 to 6 carbon positions (m0 to m6, with a color code on the side) with [$^{13}$C]-labeled atoms expressed as percentage of all corresponding molecules (MID; Mass Isotopomer Distribution). The dotted arrows corre-spond to at least two enzymatic steps. Each value corresponds to the mean of three independent experiments (biological replicates). For abbreviations, see Figs 1 and 2. The data underlying this figure can be found in https//doi.org/10.5281/zenodo.15148560.

The role of PFK in gluconeogenesis was determined by IC–MS/MS analyses of the incorporation levels of $^{13}$C atoms from metabolism of [U-$^{13}$C]-glycerol] into key gluconeogenic intermediates, in the parental, Δ*fbpase*, *RNAi*PFK (.ni and.i) and Δ*fbpase*/*RNAi*PFK (.ni and.i) cell lines (Fig 3D). Interestingly, incorporation of [$^{13}$C]-enriched atoms in H6P (F6P, M6P, G6P, and 6PG) was strongly reduced in the *RNAi*PFK.i mutant, compared to the parental and *RNAi*PFK.ni cells (Fig 3D). Unexpectedly, gluconeogenesis from glycerol was more affected in the *RNAi*PFK.i mutant than in the Δ*fbpase* mutant. Indeed, the percentage of fully [$^{13}$C]-enriched H6P (m6) from [U-$^{13}$C]-glycerol was lower in the *RNAi*PFK.i mutant than in the Δ*fbpase* mutant ($44.6 \pm 4.7\%$ *versus* $67.5 \pm 4.8\%$). In agreement with the involvement of PFK in gluconeogenesis, we also observed a synergistic effect in the Δ*fbpase*/*RNAi*PFK.i double mutants, since incorporation of [$^{13}$C]-enriched atoms in H6P was further reduced in the double mutant compared to the single Δ*fbpase* mutant ($27.8 \pm 13\%$ *versus* $67.5 \pm 4.8\%$ of the H6P are fully [$^{13}$C]-enriched, respectively). In glucose-depleted conditions (−Glc), the doubling time of the *RNAi*PFK.i cell line (18.4 h), which was identical to that of the Δ*fbpase* cell line (18.4 h) [23], was increased com-pared to the parental cells (15.7 h) (Fig 3B). The doubling time was further increased in the Δ*fbpase*/*RNAi*PFK.i cell lines (32.7 h) (Fig 3B), which suggests that the absence of PFK affects growth in the absence of glucose. Taken together, these data clearly showed that PFK participates in gluconeogenesis in PCF trypanosomes incubated in the absence of glucose with a contribution at least as important as that of FBPase. The incorporation of [$^{13}$C]-enriched atoms in H6P of the Δ*fbpase*/*RNAi*PFK.i cell line (Fig 3C) and the capacity of this mutant to grow in glucose-depleted conditions (Fig 3B), could be due to residual PFK expression. However, we cannot exclude that an additional enzyme, such as SBPase, is involved in gluconeogenesis.

## SBPase is involved in gluconeogenesis with a minor contribution

To address the possible role of SBPase (Tb927.2.5800) in gluconeogenesis, either for its SBPase activity in the PPP or for its possible FBPase activity (see Fig 1A), SBPase alleles have been replaced by the blasticidin (*BSD*) and puro-mycin (*PAC*) resistance markers, respectively, to generate the Δ*sbpase* (Δ*sbpase::BSD*/Δ*sbpase::PAC*) cell line, which was checked by PCR (S2 Fig) and western blotting (Fig 4A, line 3). The [$^{13}$C]-incorporation profile after incubation with [U-$^{13}$C]-glycerol was identical between the parental and Δ*sbpase* cell lines, questioning an SBPase contribution to glu-coneogenesis (Fig 4C). In addition, Incorporation of [$^{13}$C] atoms in S7P, the product of SBPase activity, was not affected in the Δ*sbpase* cell line (Fig 4C), suggesting that the putative cycle within the non-oxidative branch of the PPP involving

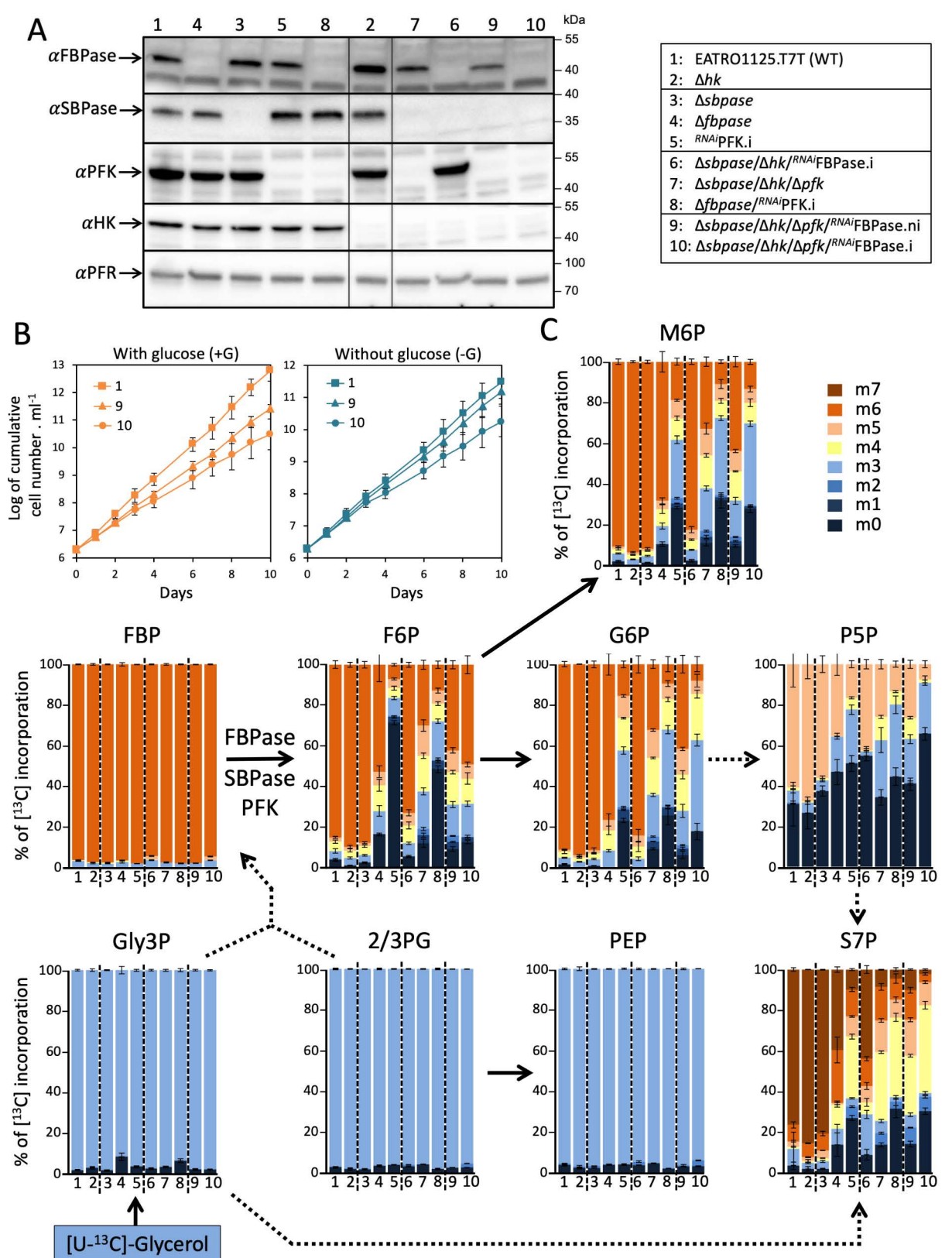

**Fig 4. IC–MS/MS analysis of intracellular metabolites extracted from FBPase, SBPase and/or PFK mutant cell lines.** Panel **A** shows a western blot analysis of all nine mutants cell lines affecting expression of FBPase, SBPase, PFK, and/or HK using the anti-FBPase, anti-SBPase, anti-PFK,

anti-HK, and anti-PFR (loading control) immune sera. For the Δ*sbpase*/Δ*hk*/Δ*pfk*/*RNAi*FBPase cell line, tetracycline-induced (.i) and non-induced (.ni) conditions have been analyzed. Lines through gels indicate where intervening lanes were cropped out of the image. Panel **B** shows growth curves of the tetracycline-induced and non-induced Δ*sbpase*/Δ*hk*/Δ*pfk*/*RNAi*FBPase cell lines and the PCF parental cells, in glucose-rich (+Glc) and glucose-depleted (−Glc) conditions. Cells were maintained in the exponential growth phase (between $10^6$ and $10^7$ cells ml$^{-1}$), and cumulative cell numbers reflect normalization for dilution during cultivation. Each value corresponds to the mean of three independent experiments (biological replicates). In panel **C**, the parasites were incubated for 2 h in PBS containing 2 mM [U-$^{13}$C]-glycerol prior to metabolite extraction. The figure shows enrichment of key gluconeogenic intermediates at 0 to 7 carbon positions (m0 to m6, with a color code on the side) with [$^{13}$C]-labeled atoms expressed as percentage of all corresponding molecules (MID; Mass Isotopomer Distribution). Each value corresponds to the mean of four independent experiments (biological replicates). The dotted arrows correspond to at least two enzymatic steps. For abbreviations, see Figs 1 and 2: P5P, Pentose 5-phosphate. The data underlying this figure can be found in https://doi.org/10.5281/zenodo.15148560.

SBPase is not significantly utilized in PCF trypanosomes for production of F6P, via the activity of transaldolase (TAL; see Fig 1A), under the conditions used.

To determine whether SBPase contributes even modestly to gluconeogenesis, it was necessary to produce an FBPase/SBPase/PFK triple mutant. This task was complicated by the lethality of the *RNAi*PFK.i cell line in glucose-rich conditions (Fig 3B). The absence of PFK certainly induces the accumulation of G6P and F6P produced by HK and then glucose-6-phosphate isomerase (PGI), respectively, at the expense of glycosomal ATP, which would rapidly become limiting and cause cell death. We can therefore expect that the deletion of the *PFK* gene in the HK null background to be viable. The *T. brucei* genomes contain two tandemly arranged *HK* genes (*HK1*, Tb927.10.2020 and *HK2*, Tb927.10.2010), which are 98.3% identical with only 11 aa difference. The *HK* genes were inactivated in the EATRO1125.T7T background using the marker-free CRISPR-Cas9 approach we recently developed for trypanosomes, by introducing in all *HK* alleles a short fragment containing stop codons and a *Bam*HI restriction site (Fig 5A) [29]. After transfection, single cells were isolated and clonal populations were screened for the insertion of repair template by PCR and *Bam*HI digestion. The PCR fragment amplified from the selected Δ*hk* cell line was fully cleaved by *Bam*HI, indicating that all *HK* alleles were inactivated by the repair cassette (Fig 5B), as confirmed by western blot analyses (Fig 5C). Abolition of glycolysis in the Δ*hk* cell line was monitored by quantifying the amounts of excreted [$^{13}$C]-enriched and non-enriched end products from metabolism of [U-$^{13}$C]-glucose and proline, respectively, using $^1$H-NMR spectroscopy (Fig 5D). Indeed, [$^{13}$C]-enriched acetate, succinate and alanine were no longer detectable in the Δ*hk* spent medium, while excretion of non-enriched acetate, succinate and alanine from the metabolism of proline remained the same as in the parental cell line. Then, the PFK alleles were successfully inactivated using the same CRISPR-Cas9 approach in the Δ*hk* background (Δ*hk*/Δ*pfk*), as confirmed by western blot analyses (Fig 5B and 5C). As expected, the doubling time of glycolysis-compromised Δ*hk* and Δ*hk*/Δ*pfk* cell lines remained the same in the proline containing PCF culture medium with or without glucose, while the parental cell line grew faster in the presence of glucose (Fig 5E).

To generate the SBPase/PFK, SBPase/FBPase and FBPase/SBPase/PFK mutants, the marker-free CRISPR-Cas9 gene inactivation and RNAi approaches were used. Briefly, the *HK* and/or *PFK* genes have been inactivated in the SBPase null background to generate the Δ*sbpase*/Δ*hk* and Δ*sbpase*/Δ*hk*/Δ*pfk* cell lines, in which the pLew100-RNAi-FBPase construct was introduced to generate the Δ*sbpase*/Δ*hk*/*RNAi*FBPase and Δ*sbpase*/Δ*hk*/Δ*pfk*/*RNAi*FBPase cell lines, respectively. The incorporation profile of [$^{13}$C]-enriched atoms in G6P, M6P, S7P, and R5P is similar in the Δ*fbpase*/*RNAi*PFK.i and Δ*sbpase*/Δ*hk*/Δ*pfk*/*RNAi*FBPase.i cell lines (Fig 4C, lanes 8 and 10, respectively), confirming that the contribution of SBPase is minor or nonexistent.

Importantly, all our attempts to generate the Δ*sbpase*/Δ*hk*/Δ*pfk*/Δ*fbpase* quadruple mutant using the CRISPR approach failed, probably because this mutant is not viable whatever the growing conditions. Indeed, if FBPase, PFK and SBPase are the only ways to produce F6P from F1,6BP, the Δ*sbpase*/Δ*hk*/Δ*pfk*/Δ*fbpase* cell line should not be able to produce the essential metabolite G6P from glucose, glycerol or proline, the main carbon sources used by the parasite. To test

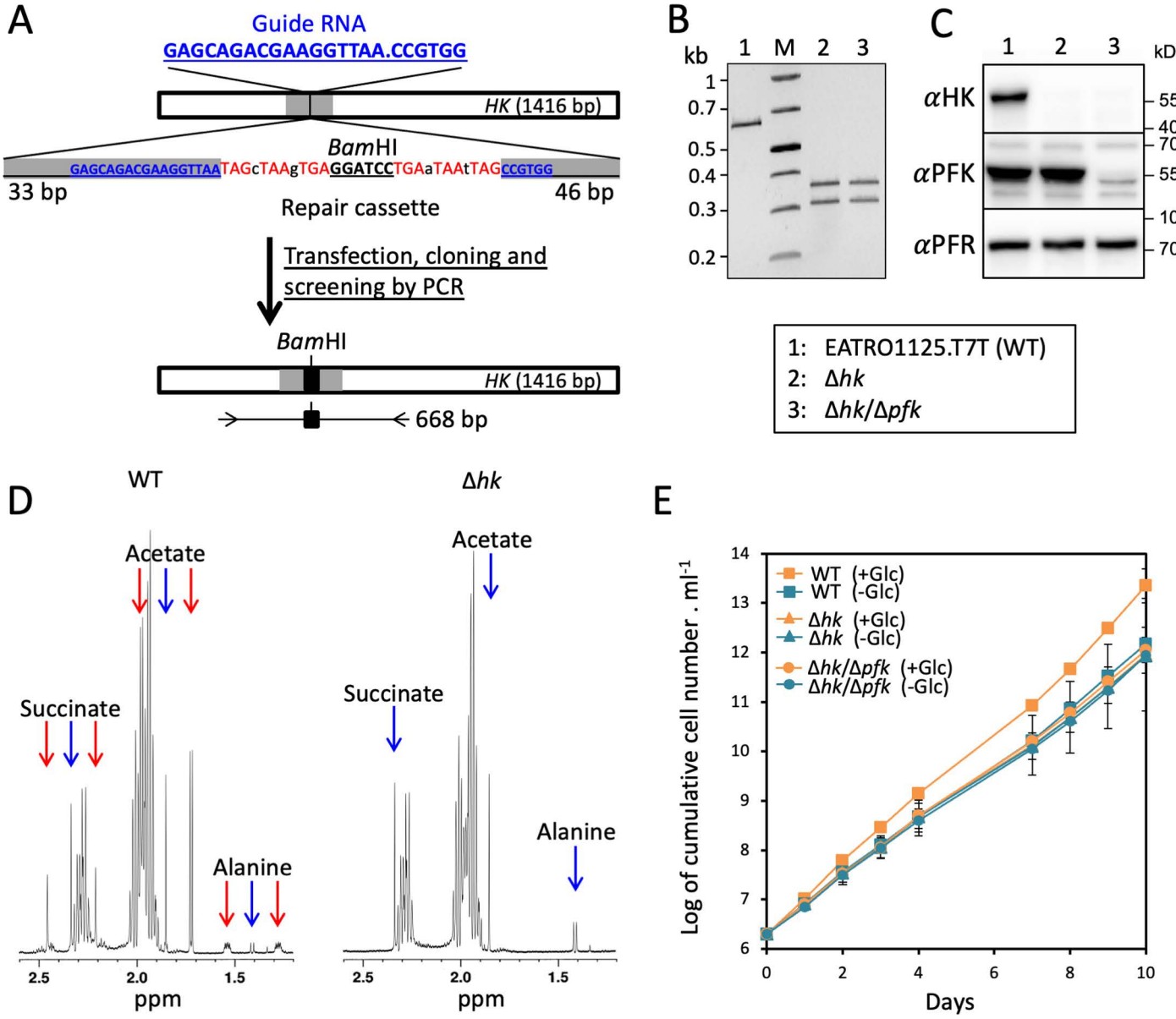

**Fig 5. HK is not essential for growth and protects against the absence of PFK.** Panel **A** shows a schematic representation of the strategy used to inactivate the *HK* genes by inserting a short sequence containing a succession of stop codons flanking a *Bam*HI restriction site. The dot in the guide RNA sequence corresponds to the Cas9 cleavage site, indicated by a vertical bar at position 509 bp in the *HK* genes. The 50-bp flanking sequences allowing repair through homology-directed repair are shown in gray in the *HK* genes and the sequence of the repair cassette. Transfection with the ribonucleoprotein complex, composed of recombinant *Streptococcus pyogenes* Cas9 (*Sp*Cas9), the *in vitro*-synthesized guide RNA and the repair cassette, results in rapid and efficient genetic modifications of the parasites, in marker-free conditions [29]. After transfection, the cells were cloned using a cell sorter in 96 well plates and the growing clonal populations were screened by PCR. The resulting edited *HK* genes and the position of the oligonucleotides used to screen clones by PCR are shown underneath. Panel **B** shows the PCR confirmation of inactivation of all *HK* alleles in the selected Δ*hk* and Δ*hk*/Δ*pfk* clones. PCR products were analyzed on agarose gel after digestion with *Bam*HI, allowing easy discrimination of gene inactivation on all alleles, by the *Bam*HI digestion of the 668-bp PCR fragment into the 359-bp and 309-bp fragments only in the Δ*hk* cell lines. Confirmation of *HK* genes inactivation by western blot analysis with the anti-HK, anti-PFK, and anti-PFR (loading control) is shown in panel **C**. Panel **D** shows $^1$H-NMR analyses of [$^{13}$C]-enriched and non-enriched end products (succinate, acetate and alanine) excreted from the metabolism of [U-$^{13}$C]-glucose and proline, respectively, by the parental (WT) and Δ*hk* cell lines incubated in PBS containing 4 mM [U-$^{13}$C]-glucose and 4 mM proline. A part of each spectrum ranging from 1.2 to 2.6 ppm is shown. The doublet resonances corresponding to [$^{13}$C]-enriched molecules produced from [U-$^{13}$C]-glucose and the single resonances corresponding to non-enriched molecules produced from proline are highlighted by red and blue arrows, respectively. Panel **E** shows growth curves of

the PCF parental, Δ*hk* and Δ*hk*/Δ*pfk* cell lines, in the proline containing PCF culture medium with also (+Glc) or without glucose (−Glc). Cells were maintained in the exponential growth phase (between $10^6$ and $10^7$ cells ml$^{-1}$), and cumulative cell numbers reflect normalization for dilution during cultivation. Each value corresponds to the mean of three independent experiments (biological replicates). The data underlying this figure can be found in https//doi.org/10.5281/zenodo.15148560.

this hypothesis, we developed an assay to quantify the efficiency of mutant generation of the *FBPase* gene, using a CRISPR-Cas9-based approach, in the parental EATRO1125.T7T, Δ*hk* and Δ*sbpase*/Δ*hk*/Δ*pfk* cell lines. After transfection with the ribonucleoprotein complex containing the guide RNA and the *Ble*R repair cassette, which confers resistance to phleomycin, cells were cloned in four 96-well plates in glycerol-rich medium supplemented with 5 µg ml$^{-1}$ phleomycin and the growing clonal populations were all screened by PCR with primers flanking the *FBPase* gene (Fig 6A). As expected, inactivation of the *FBPase* gene in the parental and Δ*hk* cell lines was highly effective, since of the 72 and 44 phleomycin-resistant clones selected, 67 and 42 had both *FBPase* alleles inactivated (homozygous clones), respectively (Fig 6B–6D). The absence of FBPase expression was confirmed by western blot analyses (Fig 6C). The few other selected clones are heterozygous, with a single inactivated *FBPase* allele, as confirmed by PCR using one primer in the *Ble*R gene and the other flanking the *FBPase* gene (Fig 6A and 6B), and western blot analyses (Fig 6C). In contrast, only five clones have been selected from the transfected Δ*sbpase*/Δ*hk*/Δ*pfk* cell line, all heterozygous, with FBPase still expressed from the active *FBPase* allele (Fig 6B–6D). Assuming that gluconeogenesis is essential in the HK null background, the failure to obtain homozygous Δ*sbpase*/Δ*hk*/Δ*pfk*/Δ*fbpase* cell lines confirmed that FBPase, PFK and also SBPase are all involved in gluconeogenesis, with no other alternative for F6P production from F1,6BP.

## FBPase expression is upregulated under glucose-depleted conditions and in the absence of PFK or SBPase

Since gluconeogenesis is required in the absence of glycolysis, the parental, Δ*fbpase*, $^{RNAi}$PFK.i, and Δ*sbpase* cell lines have been grown under glucose-depleted and glucose-rich conditions to compare the expression of FBPase, PFK, and SBPase, quantified by western blot analyses (Fig 7). FBPase expression is significantly increased by 1.4-fold in glucose-depleted conditions compared with glucose-rich conditions in the parental PCF cell line. Interestingly, this upregulation in glucose-depleted conditions was further significantly increased in the $^{RNAi}$PFK.i and Δ*sbpase* mutants (2.9- and 1.8-fold compared to WT grown in glucose-rich conditions, respectively), which is in agreement with the contribution of PFK and SBPase in gluconeogenesis. In contrast, FBPase is not upregulated in the $^{RNAi}$PFK.i and Δ*sbpase* mutants grown in glucose-rich conditions and expression of PFK and SBPase was not affected under the different conditions and in the genetic backgrounds analyzed (Fig 7).

## Glycosomal metabolite content under different carbon source conditions

In this context, it is important to determine how the glycosomal environment under gluconeogenic conditions can be understood in view of the known kinetic properties of PFK. In other words, do the gluconeogenic products of PFK (F6P and ATP) significantly drop compared to its substrates (F1,6BP and ADP) in proline-rich/glucose-free conditions? Indeed, the kinetic properties of *T. brucei* PFK imply that, to catalyze its reverse reaction, the PFK gluconeogenic substrates (F1,6BP and ADP) have to be approximately 500-fold more abundant within the glycosomes than its products (F6P and ATP) [26]. To address this question, we determined the total intracellular concentrations of metabolites by IC-HRMS by adding an internal standard ([U-$^{13}$C]-labeled *Escherichia coli* extract) into the *T. brucei* cell extracts, as described before [12]. The F1,6BP/F6P ratio was 2.5 times lower under proline conditions than under glucose conditions, whereas one would expect a high increase of this ratio to force PFK to work in the direction of gluconeogenesis in glucose-depleted conditions (Fig 8). We consider that these measured amounts reflect the glycosomal contents. Indeed, F1,6BP and F6P are primarily produced in the glycosomes and may equilibrate with the cytosolic pool through the glycosomal pores, but

**A**

Guide RNA
**CCTCTG.GACGGCAGCAGTAATAT**

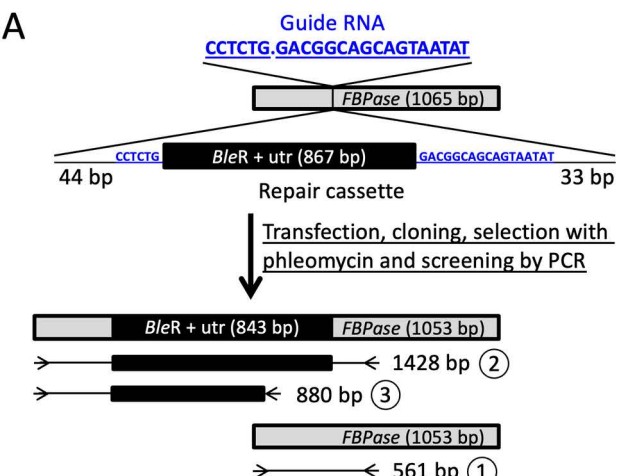

**B**

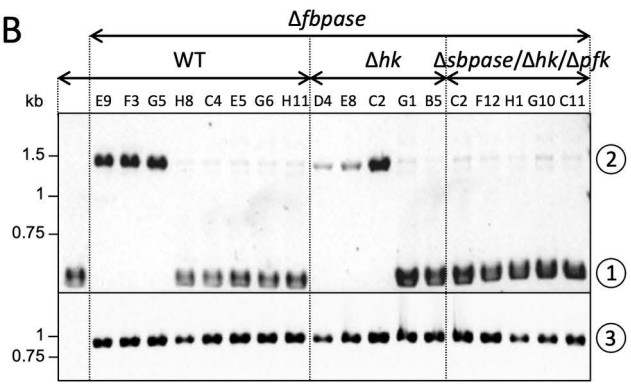

**C**

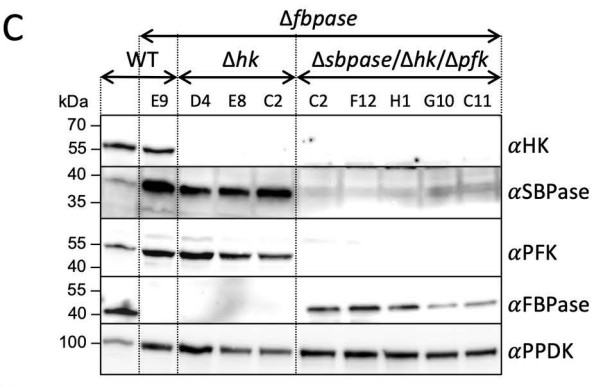

**D**

| Parental cell line | Total | *Ble*R | Homo-zygous | Hetero-zygous | % Hetero |
|---|---|---|---|---|---|
| WT | 384 | 72 | 67 | 5 | 6.9 |
| Δ*hk* | 384 | 44 | 42 | 2 | 4.5 |
| Δ*sbpase*/ Δ*hk*/Δ*pfk* | 384 | 5 | 0 | 5 | 100 |

**Fig 6. SBPase is the only possible alternative to FBPase and PFK for gluconeogenesis.** Panel **A** shows a schematic representation of the strategy used to inactivate the *FBPase* alleles by inserting the phleomycin resistance marker (*Ble*R) flanked by 5′ and 3′ untranslated regions in the *FBPase* alleles. The dot in the guide RNA sequence corresponds to the Cas9 cleavage site, indicated by a vertical bar at position 357 bp in the *FBPase* gene. After transfection of the parental EATRO1125.T7T, Δ*hk* and Δ*sbpase*/Δ*hk*/Δ*pfk* cell lines with the ribonucleoprotein complex, the cells were cloned using a cell sorter in six 96-well plates (384 wells) in the presence of 5 μg ml⁻¹ phleomycin and 10 mM glycerol before screening the growing clonal populations by PCR. The resulting edited *FBPase* gene and the position of the primers used to screen clones by PCR are shown underneath. 561-bp (1) and 1428-bp (2) PCR fragments were obtained, with primers flanking the insertion site, for the native and inactivated *FBPase* alleles, respectively. To control the correct insertion of the repair cassette, primers located in the *Ble*R gene and downstream of the insertion site were used to generate an 880-bp PCR fragment (3), which is present in all the analyzed phleomycin-resistant clones (see panel **B**). Panel **B** shows the PCR analysis of all clones containing a single inactivated allele (heterozygotes), while only three Δ*sbpase* and Δ*hk*/Δ*sbpase* clones are shown among the numerous homozygous ones (both alleles deleted) obtained for the parental and Δ*hk* transfections. PCR products obtained with the primers flanking the insertion site (1 and 2 in panel **A**) and the PCR control (3) were analyzed on agarose gel. Panel **C** confirms *FBPase* gene inactivation on one or two alleles, in selected clones, by western blot analysis with the anti-HK, anti-SBPase, anti-PFK, anti-FBPase, and anti-PPDK (loading control) immune sera. Panel **D** recapitulates for each transfection the number of plated cells (Total), of phleomycin-resistant clones (*Ble*R), of cells containing 0 (Homozygous) or 1 (Heterozygous) functional *FBPase* allele and the percentage of *Ble*R clones containing 1 functional *FBPase* allele (% Hetero). The data underlying this figure can be found in https//doi.org/10.5281/zenodo.15148560.

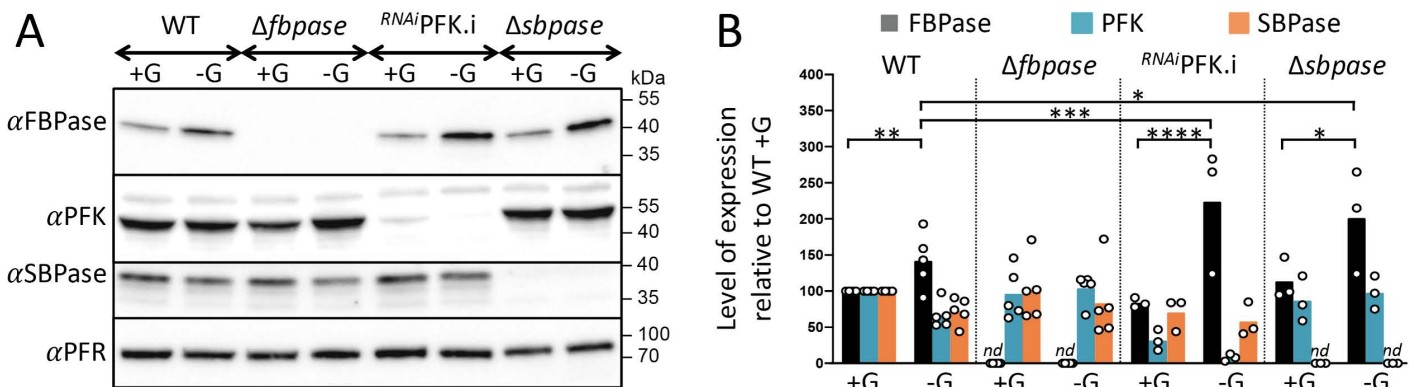

**Fig 7. Expression of FBPase, PFK, and SBPase under different growth conditions and in different genetic backgrounds.** Panel **A** shows a western blot analysis of the EATRO1125.T7T parental (WT), Δ*fbpase*, *RNAi*PFK.i, and Δ*sbpase* cell lines cultured under glucose-rich (+G) and glucose-depleted (−G) conditions during 3 days, using the anti-FBPase, anti-PFK, anti-SBPase, and anti-PFR (loading control) immune sera. In panel **B**, the corresponding specific signals after scanning the bands from 3 to 5 experiments are presented. The obtained quantitative values were normalized with the control values (PFR) and expressed as the level of expression relative to the parental cells grown under glucose-rich conditions (WT +G). Each value corresponds to the mean (±SD) of 3–5 independent experiments (biological replicates) and a Student *t* test was performed by assuming equal variances to determine the *p*-value (\**p* < 0.1; \*\**p* < 0.05; \*\*\**p* < 0.01; \*\*\*\**p* < 0.005). *nd*: not detectable. The data underlying this figure can be found in https//doi.org/10.5281/zenodo.15148560.

their possible conversion through the cytosolic glycolytic activity has probably no impact on the steady-state amounts of gluconeogenic intermediates, as previously estimated [30]. However, the observed unchanged ADP/ATP ratio is not informative. Indeed, the total intracellular amounts of ATP do not reflect the glycosomal situation, since the net production of ATP mainly occurs in the mitochondrion of the PCF trypanosomes and ATP/ADP exchange between the glycosomes and the other subcellular compartments is unlikely [30].

To estimate the impact of different carbon sources on the glycosomal ATP levels, we used an ATP-specific fluorescence resonance energy transfer (FRET)-based indicator, named ATeam, that is composed of a bacterial $F_oF_1$-ATP synthase ε subunit sandwiched between a cyan- and a yellow-fluorescent protein (CFP and YFP, respectively) [31]. In the ATP-bound form, the ε subunit retracts to bring the two fluorescent proteins close to each other, which increases FRET efficiency and allows detection of changes in ATP level upon fluorescence quantification (Fig 9A). To focus on the glycosomal ATP levels,

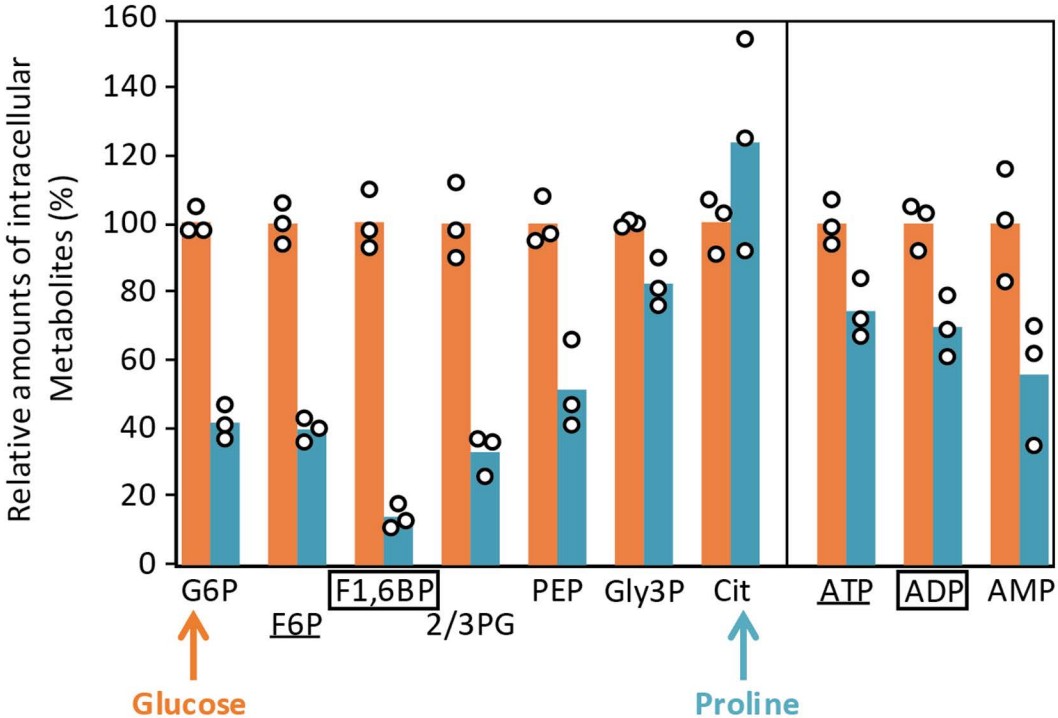

**Fig 8. Analysis of intracellular metabolites.** Intracellular concentrations of metabolites in PCF trypanosomes grown in SDM79 medium containing 10 mM glucose (Glucose) or not (Proline). The relative amounts of intracellular metabolites in proline conditions are expressed as a percentage of that in glucose conditions fixed at 100%. The sequence of the metabolic reactions is represented in linear form underneath the figure, with the entry of glucose and proline indicated by arrows. The boxed and underlined metabolites are the gluconeogenic substrates and products of PFK, respectively. Each value corresponds to the mean of three independent biological replicates. For abbreviations see Fig 1A: Cit, citrate. The data underlying this figure can be found in https//doi.org/10.5281/zenodo.15148560.

the ATeam cassette was fused to the N-terminal extremity of a Myc-tagged glycosomal protein containing a C-terminal peroxisomal targeting signal (PTS1), *i.e.,* glycerol-3-phosphate dehydrogenase (GPDH; EC 1.1.1.8; Tb927.8.3530), which was recently also used to target a cytosolic protein exclusively into this organelle [32]. The resulting ATeam-Myc-GPDH protein was conditionally expressed in the presence of tetracycline [33]. Immunofluorescence analyses showed that the ATeam-Myc-GPDH recombinant protein was located in glycosomes (Fig 9B). We previously showed that FRET efficiency was significantly reduced and the CFP fluorescence lifetime significantly increased in the $^{OE}$ATeam-Myc-GPDH.i cells incubated in the presence of glycerol compared to glucose conditions, due to accumulation of Gly3P at the expense of glycosomal ATP [33] (Fig 9C and 9D). However, FRET and CFP fluorescence lifetime were similar in proline and glucose conditions (Fig 9C and 9D), suggesting that the glycosomal ATP level is not affected by glucose depletion.

To confirm these data, we used also another ATP sensor, *i.e.,* MaLion, based on a single fluorescent protein exhibiting an increase in fluorescence emission at a single wavelength upon sensing ATP [34]. The MaLion sensor is composed of a bacterial ε subunit of $F_oF_1$-ATP synthase inserted into a fluorescent protein (Fig 9E). The conformational changes induced by the ATP-bound ε subunit increase GFP or RFP fluorescence as a function of ATP concentration. The MaLionR-Myc-GPDH recombinant protein and the MaLionG protein expressed in the $^{OE}$MaLionR-Myc-GPDH.i and $^{OE}$MaLionG.i cell lines were used to monitor glycosomal and cytosolic ATP, respectively. The glycosomal and cytosolic localization of MaLionR-Myc-GPDH and MaLionG, respectively, was confirmed by immunofluorescence analyses (Fig 9F and 9G) and tetracycline-dependent expression of MaLionR-Myc-GPDH was confirmed by western blotting using the anti-GPDH

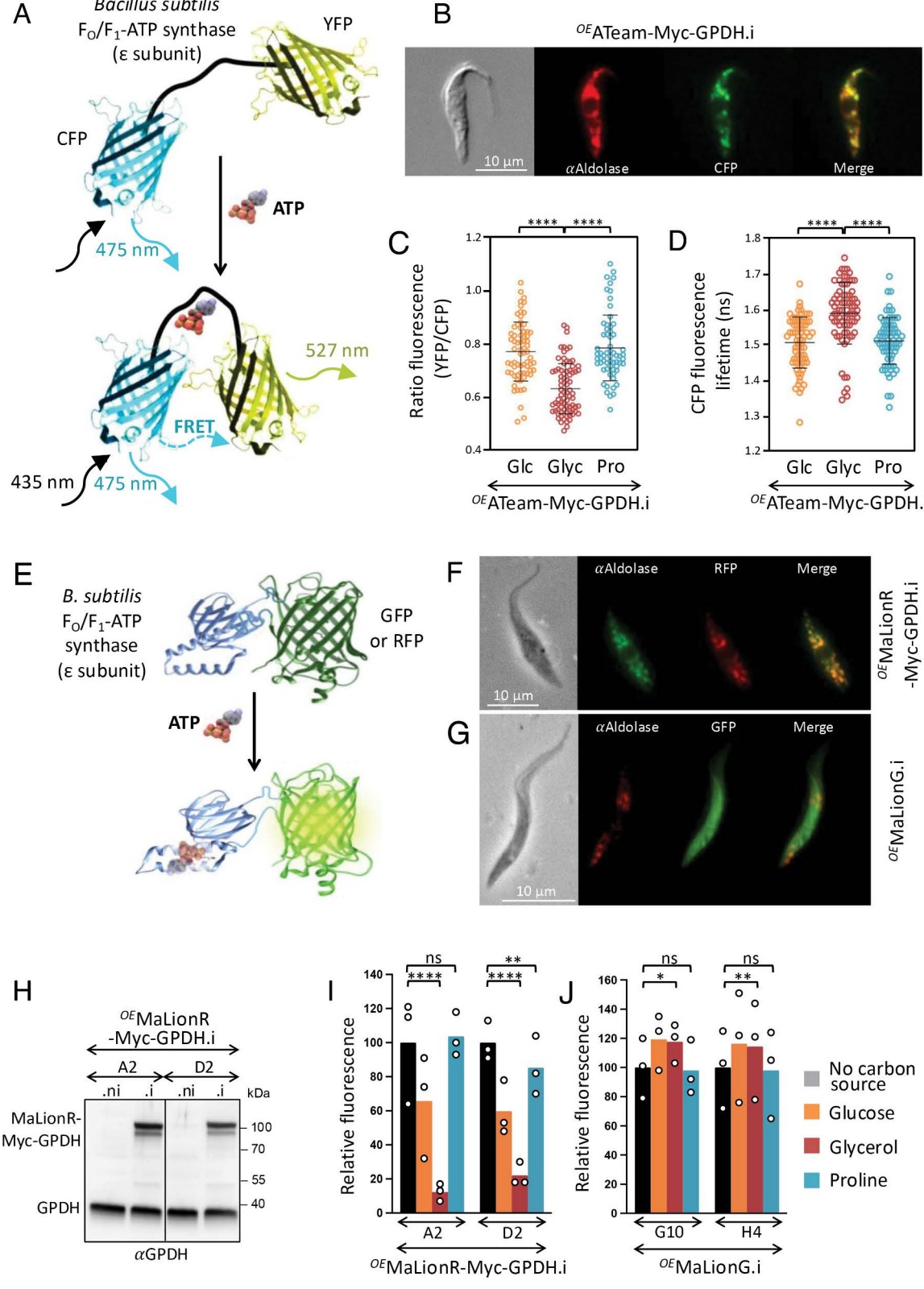

**Fig 9. Estimation of glycosomal ATP levels.** Panel **A** shows a schematic drawing of the ATeam probe adapted from [31]. Variants of CFP (mseCFP) and YFP (cp173-mVenus) were connected by the ε subunit of *Bacillus subtilis* $F_oF_1$-ATP synthase. In the ATP-free form (top), extended and flexible conformations of the ε subunit separate the two fluorescent proteins, resulting in a low FRET efficiency. In the ATP-bound form, the ε subunit retracts to draw the two fluorescent proteins close to each other, which increases FRET efficiency. In panel **B**, the glycosomal localization of ATeam-Myc-GPDH was confirmed by immunofluorescence assays on the $^{OE}$ATeam-Myc-GPDH.i cell line by using an anti-aldolase immune serum as a glycosomal marker. The yellow CFP signal was converted to green in order to merge it with the red fluorescence corresponding to aldolase. Panel **C** shows a comparative analysis of the ratio of YFP emission (FRET) and CFP emission after excitation at 435 nm in the $^{OE}$ATeam-Myc-GPDH.i cell line incubated in the presence of 5 mM of glucose (Glc), glycerol (Glyc) or proline (Pro) (means ±SD, $n=2$ independent experiments, ****$p$-value associated with the Student $t$ test is < 0.0001). Panel **D** shows a comparative analysis of the CFP fluorescence lifetime of the same cell line incubated in the same conditions (means ±SD, $n=3$ independent experiments, ****$p$-value associated with the Student $t$ *test* is < 0.0001). Panel **E** shows a schematic drawing of the MaLion probe adapted from [34]. The ε subunit of *B. subtilis* $F_oF_1$-ATP synthase was inserted into the green fluorescent protein (GFP) and the red fluorescent mApple protein (RFP) to design the single fluorescence protein-based ATP indicators, named MaLionG and MaLionR, respectively. In the ATP-bound form, the ε subunit retracts and induces a conformational change of the GFP and RFP, which becomes fluorescent. Panels **F** and **G** show the glycosomal and cytosolic localization of MaLionR-Myc-GPDH and MaLionG by immunofluorescence assays on the $^{OE}$MaLionR-Myc-GPDH.i and $^{OE}$MaLionG cell lines, respectively, by using an anti-aldolase immune serum as a glycosomal marker. Expression of MaLionR-Myc-GPDH in the tetracycline-induced cell line (.i) was confirmed in Panel **H** by western blotting using the anti-GPDH immune serum. Panels **I** and **J** shows fluorescence analyses of the $^{OE}$MaLionR-Myc-GPDH.i and $^{OE}$MaLionG.i mutants (two cell lines for each), respectively, maintained during several days in glucose-depleted conditions, washed and then incubated during 30 min in PBS containing or not (No carbon source) 5 mM of glucose, glycerol or proline before microscopic analyses (means ± SD, $n=3$ independent experiments and a Student $t$ tes*t* was performed by assuming equal variances to determine the $p$-value, *$p < 0.1$; **$p < 0.05$; ***$p < 0.01$; ****$p < 0.005$; ns, not significant). The obtained quantitative values were expressed as the level of expression relative to the "No carbon source" condition. The data underlying this figure can be found in https//doi.org/10.5281/zenodo.15148560.

immune serum (Fig 9F). Fluorescence emission was measured in cells incubated in PBS containing glucose, glycerol, proline or no carbon source. The fluorescence intensity, which reflects the amounts of ATP, is not affected by the absence of carbon source compared with glucose conditions. This is probably due to the maintenance of the cellular ADP/ATP ratio when an unknown endogenous carbon source is used in the absence of external carbon sources [27], with a considerable reduction in metabolic fluxes. As observed with the $^{OE}$ATeam-Myc-GPDH.i cell line, two $^{OE}$MaLionR-Myc-GPDH.i cell lines showed a strong reduction of glycosomal ATP under glycerol conditions, while incubation with proline induced a significant increase in glycosomal ATP compared with glucose conditions (Fig 9I). However, as expected, cytosolic ATP levels remained relatively stable whatever the incubation conditions (Fig 9J). Taken together, these surprising data suggest that PFK converts F1,6BP to F6P in the presence of relatively high levels of ATP, which is not in agreement with what is predicted to occur free in solution, with the known equilibrium constant of the PFK reaction and the enzyme's known kinetic properties [35].

## Discussion

*In vivo*, procyclic trypanosomes have adapted their metabolism to the insect midgut where glucose is rare or absent and amino acids, such as proline, are predominant. As a result, they have developed a proline-based catabolism that takes place mainly in the mitochondrion, where ATP is mainly produced by oxidative phosphorylation. This metabolic strategy adapted to the trypanosome's natural environment makes glycolysis unnecessary, however, the glycolytic enzymes are required to work in reverse to supply several essential metabolic pathways with G6P. It is noteworthy that the possible presence of glycerol in the insect midgut would significantly stimulate gluconeogenesis, however, no *in vivo* data are yet available to validate this hypothesis. A special step in gluconeogenesis is the production of F6P from F1,6BP by FBPase, since the reaction by the glycolytic enzyme ATP-dependent PFK is considered irreversible *in cellulo*. We showed here that PCF trypanosomes use three different enzymes to catalyze this reaction, *i.e.* SBPase and the ATP-dependent PFK in addition to the canonical gluconeogenic FBPase.

The contribution of SBPase, which is marginal compared with that of FBPase and PFK, is not surprising since members of this protein family, such as Shb17 in yeast and F-I in cyanobacteria, can dephosphorylate both F1,6BP (FBPase activity) and S1,7BP (SBPase activity) [19,20]. In agreement with the contribution of SBPase to the glycosomal FBPase

activity, we previously reported that the glycosomal FBPase activity was reduced, but not abolished, in the Δ*fbpase* mutant cell line [23]. The marginal contribution of SBPase makes the cycle in the non-oxidative branch of the PPP, which produces F6P from G3P and DHAP (see Fig 1A), unlikely in these growth conditions. This pathway, proposed by Hannaert and colleagues in an attempt to integrate SBPase into the metabolism of trypanosomes [18], before its validation in yeast [19], has not yet been described in trypanosomatids [36].

More surprisingly, PFK contributes at least as much as FBPase to F6P production, as evidenced by (*i*) the strong reduction of H6P production from F1,6BP in the ^RNAiPFK.i cell line, even more than in the absence of FBPase (Δ*fbpase* mutant) or FBPase/SBPase (Δ*fbpase*/Δ*hk*/^RNAiFBPase.i mutant), (*ii*) the significant production of F6P from F1,6BP in the absence of FBPase and SBPase (Δ*fbpase*/Δ*hk*/^RNAiFBPase.i mutant), confirming that the only alternative for F6P production, *i.e.*, PFK, contributes greatly to gluconeogenesis, and (*iii*) the significant increase in FBPase expression in glucose-depleted conditions, in the absence of PFK, probably to compensate for the PFK-dependent reduction in F6P production. Fernandes and colleagues recently published the first demonstration that the recombinant human and trypanosomatid ATP-dependent PFKs, purified from *E. coli*, can *in vitro* catalyze the production of F6P from F1,6BP, if the product of the relative concentrations of their gluconeogenic substrates (F1,6BP and ADP) is at least 500-fold greater than that of their products (F6P and ATP) [26], however, no data to date have validated this possibility *in cellulo*. Our data therefore provide the first evidence that ATP-dependent PFKs can operate *in vivo* in the direction of gluconeogenesis, even more surprisingly in the presence of FBPase activity and with no apparent imbalance in the PFK's products/substrates ratio. These unexpected data raise a number of questions such as, how does the ATP-dependent PFK carry out the reverse reaction in an unfavorable metabolic environment? What is the added value of PFK compared with FBPase in glucose-depleted conditions? Do these data shed new light on the driving force behind the sequestration of glycolysis in the ancestors of glycosome-bearing organisms (kinetoplastids, which include trypanosomatids, and diplonemids)? Does the specific PFK behavior apply to, and impact the metabolism of, the glucose-dependent BSF trypanosomes?

Nearly 50 years after the discovery of glycosomes [9], the selective advantage of compartmentalization of glycolytic enzymes in kinetoplastid/diplonemid ancestors is still being debated [11]. The glycosome has an electron-dense granular matrix and sometimes displays in BSF a crystalloid core in electron-microscopic analysis [37,38], suggesting that the glycolytic enzymes, which account for up to 90% of the glycosome's total protein [39], have a higher degree of organization possibly compatible with metabolic channeling [40,41]. Metabolic channeling is a process by which intermediates in a metabolic pathway are directly transferred between enzymes without diffusing into the surrounding solution, enhancing the efficiency and regulation of biochemical reactions [42]. The initial obvious idea was that glycosomes provide the possibility to speed up the glycolytic flux, in particular through metabolic channeling. However, a number of arguments make this hypothesis unlikely. First, the highest glycolytic flux reported in trypanosomatids, described for the BSF *T. brucei*, is not exceptionally high, since other cells may sustain a considerably higher glucose consumption rate without any form of glycolytic enzyme compartmentalization [43]. Second, simulations using a mathematical kinetic model of glycolysis in BSF *T. brucei* correctly reproduced the steady-state process, and predicted fluxes and metabolite concentrations that had previously been determined under various conditions [44]. However, removal of the glycosomal membrane in the model, allowing the enzymes, cofactors and intermediates to distribute over the entire cytosol, had little effect on the steady-state flux [45]. Third, the catalytic rates of glycolytic enzymes in a cross-linked complex of *T. brucei* did not differ from those of the same enzymes in solution, which confirmed that glycosomal sequestration of glycolytic enzymes does not stimulate glycolysis [46]. However, the possibility that gluconeogenesis could benefit from this sequestration has not yet been studied. One may consider that PFK and its flanking enzymes in the glycolytic/gluconeogenic pathway (aldolase and PGI, respectively) form a glycosomal metabolon channeling the substrates/products, that could enable the thermodynamically unfavorable reverse PFK enzymatic reaction. Such thermodynamically unfavorable reactions have been described as being facilitated through metabolic channeling, by increasing local concentrations

of substrates and products, helping to overcome thermodynamic barriers that would be unfavorable in a more dilute or non-channeled environment. Such mechanisms have been observed in pathways like glycolysis, the TCA cycle, and other metabolic processes where channeling can stabilize intermediates and drive forward reactions that would otherwise be less favorable [47]. This hypothesis is also consistent with the observation that mild detergent treatments of glycosome extracts release particle-bound complexes containing eight glycolytic enzymes, including PFK, aldolase and PGI [40]. An alternative to the metabolic channeling hypothesis would be the existence of subpopulations of glycosomes, as recently described for fungal peroxisomes [48], dedicated to glycolysis or gluconeogenesis depending on their protein content. In this hypothetical model, low abundance "gluconeogenic glycosomes" would contain enzymes and metabolic pathways compatible with a drastic reduction in ATP and F6P, as opposed to the more abundant "glycolytic glycosomes". However, this hypothesis of glycosomal heterogeneity already described between different PCF cells, has not yet been reported within the same trypanosome cell [49].

The use of PFK for gluconeogenesis may be helpful to solve the question of the potential futile cycle involving FBPase and the ATP-dependent PFK, which consists of alternating synthesis and hydrolysis of F1,6BP, where FBPase converts F1,6BP to F6P, and PFK reverses this reaction, leading to a wasteful consumption of ATP. Indeed, since PFK and FBPase are constitutively co-expressed in the PCF trypanosomes regardless of growth conditions (see Fig 7), a mechanism preventing this futile cycle is required to preserve the glycosomal ATP pool essential for glycosomal anabolic pathways [50]. Preservation of the glycosomal ATP is also important for gluconeogenesis from proline or glycerol, since the first glycosomal enzymatic steps of proline and glycerol catabolism require ATP (for phosphoenolpyruvate kinase (PEPCK)/pyruvate phosphate dikinase (PPDK) and glycerol kinase (GK), respectively) [23]. However, how PFK activity is regulated, in particular in the natural gluconeogenic environment of the PCF trypanosomes, is currently not understood. For instance, PFK from *T. brucei* is not allosterically inhibited by high concentrations of ATP, unlike other PFKs, as exemplified by PFK2 from *E. coli* for which expression of mutant no longer inhibited by ATP showed increased PFK activity during gluconeogenesis [51]. The only known metabolic effector of the *T. brucei* enzyme is AMP, which acts as a moderate activator and does not appear to be relevant in preventing futile cycling under glucose-depleted conditions [17]. Thus, PFK operating in the same gluconeogenic direction than FBPase under glucose-depleted conditions, is certainly an added value for the trypanosomes to avoid uncontrolled waste of glycosomal ATP.

Maintaining an ATP/ADP balance within glycosomes is quite challenging for trypanosomes, since it is generally accepted that no exchange of nucleoside phosphates takes place between the glycosomal and cytosolic compartments. This implies that each ATP molecule consumed within the organelle must be regenerated within the organelle [52]. As mentioned above, the first glycosomal steps of gluconeogenesis from proline and glycerol (PEPCK, PPDK, and GK) invests ATP molecules. However, ATP is not regenerated through the conventional downstream gluconeogenic enzymatic steps, since FBPase wastes the high-energy phosphate bond of F1,6BP by releasing Pi. A gluconeogenic PFK-producing ATP makes it possible to maintain this ATP/ADP glycosomal balance, if PFK converts all F1,6BP to F6P. Since glycosomal FBPase activity probably also contributes to F6P production, PCF grown under glucose-depleted conditions may have developed other means to produce ATP in the organelle. The most obvious possibility is the production of glycerol by GK from catabolism of proline, as well as production of malate or pyruvate by PEPCK or PPDK, respectively, from catabolism of glycerol [23]. In addition, *T. brucei* expresses glycosomal and cytosolic arginine kinases, which could create an energy-buffering/relay system by exchanging high-energy phosphate bonds between these two subcellular compartments [53]. However, this putative phosphoarginine energy buffering system must be tightly controlled to prevent the possibly lethal homogenization of the glycosomal and cytosolic ATP pools, particularly in glucose-rich conditions. Indeed, the lack of regulation of trypanosome HK and PFK activity by the products of their reactions implies a risk of unrestricted accumulation of glycolytic intermediates because of the so-called "turbo design" of the glycolytic pathway, *i.e.,* the principle of investing ATP at the beginning of the process with a yield of surplus ATP after its completion, which will further stimulate the two kinases at the beginning of the pathway (*therefore it is called 'turbo-design'*) beyond the capacity of

the downstream enzymes, thus leading to uncontrolled accumulation of the intermediates. To avoid the lethal potential of uncontrolled trypanosome glycolysis, no significant exchange of ATP should occur between the glycosomes and other subcellular compartments.

Interestingly, the trypanosomatid ATP-dependent PFK belongs to a group consisting predominantly of PPi-dependent PFK, suggesting that the trypanosome ancestor expressed a PPi-dependent PFK with a possible gluconeogenic role, which changed its phospho-donor specificity [16]. However, a recent phylogenetic analysis of PPi-dependent and ADP-dependent PFK from organisms belonging to phylum Euglenozoa, including Kinetoplastidea, Diplonemea and Euglenida classes, suggested that PFK ('clade X') of the common Euglenozoa ancestor, retained in the trypanosomatid lineages, was almost certainly ATP dependent, indicating that phospho-donor specificity had already changed from PPi to ATP well before the ancestral euglenozoan [54]. Consequently, the trypanosomatid ancestor probably never expressed a PPi-dependent glycosomal PFK, since glycosomes (*i.e.,* peroxisomes having acquired glycolytic/gluconeogenic enzymes) are only found in kinetetoplastids and diplonemids, not euglenids, thus must have appeared in the Euglenozoa class [54] only after the transition from PPi to ATP dependence.

BSF trypanosomes live in the fluids as well as in the interstitial spaces of the tissues of their mammalian hosts, where glucose is present at homeostatic concentrations ranging from 5 to 4 mM, respectively. Consequently, they do not require gluconeogenesis to produce the essential G6P, which is permanently available from glycolysis. Thus, the constitutive expression of FBPase and SBPase in BSF raises questions about their role in glucose-rich conditions. Kovarova and colleagues, successfully inactivated the *FBPase* and *SBPase* genes without affecting gluconeogenesis capacity, suggesting that PFK could also be involved in gluconeogenesis in BSF [36]. However, the authors considered this possibility unlikely based on the fact that the PFK could be depleted by 70% without effect on gluconeogenesis and growth [36]. These data can also be interpreted as a consequence of a large excess of PFK activity, which remained high enough to sustain the same gluconeogenic flux when its expression was reduced ~4 times. The much higher expression of PFK in BSF compared to PCF, while FBPase and SBPase expression is of the same order in both parasitic forms (Fig 1B), would certainly support this alternative interpretation of the data. In any case, the gluconeogenic activity of PFK is probably irrelevant for BSF trypanosomes since the parasite never faces glucose-depleted environments *in vivo*. This suggests that FBPase has another unknown function in BSF under glucose-rich conditions, as previously proposed for PCF [25], that deserves to be investigated. One possibility is the production of Pi by FBPase activity, which is used to feed GAPDH, another glycosomal enzymatic activity downstream in glycolysis producing 1,3-bisphosphoglycerate from G3P. In vertebrates, FBPase is considered to be a multifaceted cellular regulator [55], which opens up new perspectives for addressing this question.

## Materials and methods

### Trypanosomes and cell cultures

The PCF of *T. brucei* EATRO1125.T7T (TetR-HYG T7RNAPOL-NEO), which constitutively expresses the T7 RNA polymerase gene and the tetracycline repressor under the control of a T7 RNA polymerase promoter for tetracycline inducible expression [56], was cultivated at 27 °C in the presence of 5% $CO_2$ in SDM79 medium containing 10% (v/v) heat-inactivated fetal calf serum and 3.5 mg ml$^{-1}$ hemin [57]. Alternatively, a glucose-depleted medium derived from SDM79, named SDM79-GlcFree, was developed to cultivate the procyclic cells [23]. This SDM79-GlcFree medium consists of glucose-depleted SDM79 medium, containing 20% (v/v) heat-inactivated fetal calf serum, in which the EATRO1125.T7T cells were cultured during 72h to consume the glucose coming from the serum and then diluted with the same volume of glucose-depleted SDM79 medium without serum to finally obtain SDM79-GlcFree. Glucose depletion was verified by NMR spectrometry analyses (with a detection threshold ≤1 μM) and to prevent import of residual glucose, 50 mM *N*-acetyl-D-glucosamine, a non-metabolized glucose analogue inhibiting glucose transport, were added to the medium. The glucose-rich and glycerol-rich conditions were obtained by adding to SDM79-GlcFree 10 mM glucose (without *N*-acetyl-D-glucosamine) or 10 mM glycerol, respectively.

## Gene knockout

Generation of the Δ*fbpase* (EC 3.1.3.11; Tb927.9.8720) null mutant and rescue Δ*fbpase*/FBPase cell line was described before [23]. Replacement of both alleles of the *SBPase* gene (EC 3.1.3.11; Tb927.2.5800) by the BSD and PAC resistance markers *via* homologous recombination was performed with DNA fragments containing a resistance marker gene flanked by the SBPase UTR sequences. Briefly, the *BSD* or *PAC* orf were PCR amplified with primers containing 50-bp upstream (5′ primer) or downstream (3′ primer) of the ATG or stop codons, respectively, of the *SBPase* gene. The EATRO1125.T7T parental cell line was used to generate the SBPase knockout cell line (Δ*sbpase::BSD*/Δ*sbpase*::*PAC*, called Δ*sbpase*). After transfection, selection was done in SDM79 medium containing hygromycin (25 µg ml$^{-1}$), neomycin (10 µg ml$^{-1}$), blasticidin (10 µg ml$^{-1}$), and/or puromycin (1 µg ml$^{-1}$).

## Gene inactivation by CRISPR/Cas9 inactivation

Gene inactivation was achieved by inserting double-stranded DNA (repair cassette) corresponding either to a short sequence (TAGCTAAGTGA*ggatcc*TGAATAATTAG) containing six successive stop codons in the three reading phases (underlined in the short sequence above) and a *Bam*HI restriction site (italic letters in the short sequence above) or to a resistance marker (phleomycin, *Ble*R), as recently described [29]. The repair cassettes were also flanked by 50-bp homologous to the 5′ and 3′ sequences of the Cas9 cut site. To inactivate the *HK* (EC 2.7.1.1; Tb927.10.2010 and Tb927.10.2020), *PFK* and *FBPase* genes, the RNA guides used are GAGCAGACGAAGGTTAACCGTGG (ARNg-HK_134), GTTGATACTCTCGAGCGCCCGG (ARNg-PFK_94), and ATATTACTGCTGCCGTCCAGAGG (ARNg-FBPase-374_rev), respectively. Prior to transfection, equal quantities (0.4 µmol each) of TracrRNA and gRNA from Integrated DNA Technologies (IDT) were combined. The mixture was then heated at 95 °C for 5 min and gradually cooled to room temperature. Subsequently, this mixture was combined with 30 µg of *Sp*Cas9 sourced from IDT (Alt-RTM *Sp*Cas9 Nuclease 3NLS, Catalog #1074181). 5 × 10$^5$ cells were respectively transfected using Amaxa nucleofector device with Basic Parasite Nucleofector Kit 2 (Lonza) (100 µl) with 1 µg of purified repair cassette (in a final volume of 10 µl) and 30 µg of Cas9 protein preloaded with the mixture of TracrRNA and gRNA. Cells were transfected using program X-001 and resuspended in 5 ml of medium containing or not 5 µg ml$^{-1}$ phleomycin. Following a 2-day culture period, a portion of the transfected cells underwent DNA extraction using the NucleoSpin Blood kit (Macherey-Nagel) followed by PCR amplification using primers that flank the Cas9 cleavage site. Gel analysis of the PCR products is expected to show two bands corresponding to the wild-type and modified gene. Cells were subsequently cloned using a cell sorter (TBM Core facility), and the selection of inactivated cells was performed by DNA extraction using the NucleoSpin Blood kit (Macherey-Nagel) followed by PCR amplification using primers flanking the Cas9 cleavage site as previously described. Guide RNAs were designed using EuPaGDT [58], from http://tritrypdb.org. Primers and guide RNAs used were synthesized by IDT. All genetic modifications were confirmed by sequencing the PCR product (Eurofins) generated from primers flanking the modifications.

## Inhibition of gene expression by RNAi

RNAi-mediated inhibition of *FBPase* gene expression was performed in the TetR-HYG and T7RNAPOL-NEO background by expression of stem-loop "sense-antisense" RNA molecules of the targeted sequences [56,59] using the pLew100 expression vector, which contains the phleomycin resistance gene (kindly provided by E. Wirtz and G. Cross) [60]. Production of the pLew-FBPase-SAS plasmid and the $^{RNAi}$FBPase cell lines was described before [23]. To generate the Δ*sbpase*/Δ*hk*/$^{RNAi}$FBPase and Δ*sbpase*/Δ*hk*/Δ*PFK*/$^{RNAi}$FBPase mutants the Δ*sbpase*/Δ*hk* and Δ*sbpase*/Δ*hk*/Δ*pfk* cell lines were transfected with the *Not*I-linearized pLew100-FBPase-SAS plasmid in 2-mm electroporation cuvettes with Basic Parasite Nucleofector Kit 2 on the Amaxa Nucleofector device (Lonza), using X-001 program. Selection of the mutant cell lines was performed in glucose-rich SDM79 medium containing hygromycin (25 µg ml$^{-1}$), neomycin (10 µg ml$^{-1}$), puromycin (1 µg ml$^{-1}$), blasticidin (10 µg ml$^{-1}$), and phleomycin (5 µg ml$^{-1}$).

To down-regulate expression of the *PFK* gene (Tb927.3.3270) by RNAi, we produced the pLew-PFK-SAS plasmid, which is the pLew100 vector containing a sense and antisense version of the targeted 736-bp fragment (from position 578 to 1314 of the *PFK* gene), separated by a 67-bp fragment, under the control of a PARP promoter linked to a prokaryotic tetracycline operator. Briefly, a PCR-amplified 775-bp fragment, containing the antisense *PFK* sequence with restriction sites added to the primers, was inserted into the *Hind*III and *Bam*HI restriction sites of the pLew100 plasmid. A 736-bp PCR-amplified fragment containing the sense *PFK* sequence was then inserted upstream of the antisense sequence, using *Hind*III and *Xho*I restriction sites (*Xho*I was introduced at the 3′-extremity of the antisense PCR fragment). After transfection of the EATRO1125.T7T and Δ*fbpase* cell lines with the *Not*I-linearized pLew-PFK-SAS plasmid, as described above, selection of the $^{RNAi}$PFK and Δ*fbpase*/$^{RNAi}$PFK mutants was performed in glucose-rich SDM79 medium containing hygromycin (25 µg ml$^{-1}$), neomycin (10 µg ml$^{-1}$), and phleomycin (5 µg ml$^{-1}$), plus puromycin (1 µg ml$^{-1}$) and blasticidin (10 µg ml$^{-1}$) for the Δ*fbpase*/$^{RNAi}$PFK cell lines. Aliquots of all selected mutant cell lines were frozen in liquid nitrogen to provide stocks of each line that had not been cultivated long term in medium. Induction of RNAi cell lines was performed by addition of 10 µg ml$^{-1}$ tetracycline.

### Expression of ATeam and MaLion recombinant proteins

To express in glycosomes the ATeam cassette, composed of the ε subunit of the $F_oF_1$-ATP synthase sandwiched by the CFP and YFP [31], the full-length *GPDH* gene (Tb927.8.3530) encoding the PTS1-containing glycosomal glycerol-3-phosphate dehydrogenase was inserted downstream of the ATeam cassette to generate a gene encoding the glycosomal ATeam-Myc-GPDH recombinant protein, as described before [33]. Briefly, a PCR-amplified 1833-bp fragment containing the ATeam sequence flanked by the *Hind*III and *Nde*I restriction sites, respectively, was inserted into the *Hind*III and *Nde*I restriction sites of the pLew100 vector containing the *GPDH* gene preceded by 10 Myc tag sequences. The resulting pLew-ATeam-Myc-GPDH plasmid was introduced in the EATRO1125.T7T parental cell line and clones were selected with phleomycin (5 µg ml$^{-1}$), in addition to hygromycin and neomycin.

To generate the pLew-MaLionR-Myc-GPDH plasmid, the ATeam cassette (*Hind*III-*Nde*I fragment) of the pLew-ATeam-Myc-GPDH plasmid was replaced by a PCR fragment containing *MaLionR* gene flanked by the *Hind*III and *Nde*I restriction sites, PCR amplified from the MaLionR plasmid (#113908, Addgene). To generate the pLew-MaLionG plasmid, a PCR fragment containing *MaLionG* gene (#113906, Addgene) flanked by the *Hind*III and *Bam*HI restriction sites was inserted in the *Hind*III/*Bam*HI sites of pLew100. The resulting pLew-MaLionR-Myc-GPDH and pLew-MaLionG plasmids were introduced in the EATRO1125.T7T parental cell line and clones were selected with phleomycin (5 µg ml$^{-1}$), in addition to hygromycin and neomycin.

### Western blot analyses

Total protein extracts of PCF of *T. brucei* ($5 \times 10^6$ cells) were separated by SDS-PAGE (10%) and immunoblotted on TransBlot Turbo Midi-size PVFD Membranes (BioRad) [61]. Immunodetection was performed as described [61,62] using as primary antibodies, the rabbit anti-FBPase (1:1000), the rabbit anti-SBPase (1:250), the rabbit anti-PFK (1:200), the rabbit anti-PFR (1:10,000), the rabbit anti-HK (1:5000), the rabbit anti-GPDH (1:1000) [63], the rabbit anti-PPDK (1:1000) [64], and mouse anti-Myc 9E10 (αMyc; 1:100; gift from K. Ersfeld, Hull, UK). Anti-rabbit or anti-mouse IgG conjugated to horseradish peroxidase (Bio-Rad, 1:5000 dilution) was used as secondary antibody. Revelation was performed using the Clarity Western ECL Substrate as described by the manufacturer (Bio-Rad). Images were acquired and analyzed with the ImageQuant LAS 4000 luminescent image analyzer.

### NMR spectrometry experiments

$2 \times 10^7$ procyclic *T. brucei* were centrifuged at 1400*g* for 10 min, then the pellet was washed twice with PBS and the cells were incubated for 6 h at 27 °C in 1 ml of incubation buffer (PBS supplemented with 5 g l$^{-1}$ NaHCO$_3$, pH 7.4) with 4 mM

[U-$^{13}$C]-glucose and with or without 4 mM of non-enriched proline. The viability of the cells during the incubation was checked by microscopic observation. At the end of the incubation, 500 μl supernatant were collected and 50 μl of maleate solution in D$_2$O (10 mM) was added as internal reference. $^1$H-NMR spectra were performed at 500.19 MHz on a Bruker Avance III 500 HD spectrometer equipped with a 5 mm cryoprobe Prodigy. Measurements were recorded at 25 °C. Acquisition conditions were as follows: 90° flip angle, 5,000 Hz spectral width, 32 K memory size, and 9.3 s total recycle time. Measurements were performed with 64 scans for a total time close to 10 min 30 s. Resonances of the obtained spectra were integrated and metabolites concentrations were calculated using the ERETIC2 NMR quantification Bruker program [65]. To quantify glucose-derived acetate and succinate, we used the following procedure. Protons linked to acetate carbon C2 generates by $^1$H-NMR five resonances, a single peak ([$^{12}$C]-acetate) flanked by two doublets ([$^{13}$C]-acetate). When [U-$^{13}$C]-glucose is the only carbon source, the central resonance (1.85 ppm) corresponding to [$^{12}$C]-acetate, probably derived from an unknown internal carbon source. As a consequence, it is not included in the glucose-derived acetate quantitative analyses.

## Mass spectrometry analyses of [$^{13}$C]-incorporation into cellular metabolites

For analysis of [$^{13}$C]-incorporation into intracellular metabolites, EATRO1125.T7T parental and mutant cell lines grown in SDM79 medium were washed twice with PBS and resuspended in incubation solution (PBS containing 2 mM [U-$^{13}$C]-glycerol or 2 mM [U-$^{13}$C]-proline). The cells were incubated for 2 h at 27 °C before being collected on filters by fast filtration preparation for mass spectrometry analysis, as described before [12]. Total sampling time was below 8 s and the extraction of intracellular metabolites was carried out by transferring the filters containing the pellets into 5 ml of boiling water for 30 s. The extracts were briefly vortexed (~2 s), immediately filtered (0.2 μm), and chilled with liquid nitrogen. After lyophilization, the dried extracts were resuspended in 200 μl Milli-Q water prior to analysis. Three replicates were taken from each culture media, sampled and analyzed separately. Metabolites were analyzed by ionic-exchange chromatography coupled with tandem mass spectrometry (IC–MS/MS) using the method described by Bolten and colleagues [66]. Retention time on the column and multiple reactions monitoring transition of each analyzed metabolite are shown in Table 1 of [67]. The $^{13}$C mass isotopomer distribution of intracellular metabolites was determined from relevant isotopic clusters in the IC–MS/MS analysis, according to Kiefer and colleagues [68]. $^{13}$C mass isotopomer distribution measurements were performed using a triple quadrupole mass spectrometer (4000Qtrap, Applied Biosystems). To obtain $^{13}$C-labeling patterns ($^{13}$C isotopologues), isotopic clusters were corrected for the natural abundance of isotopes other than $^{13}$C, using the in-house software IsoCor (available at MetaSys) [69].

## Determination of intracellular metabolite concentrations by IC-HRMS and enzymatic assays

For the quantification of intracellular metabolites, the *T. brucei* PCF EATRO1125.T7T cell line maintained 2 h in PBS containing 10 mM glucose or 10 mM proline was sampled by fast filtration, as described above. The extraction of intracellular metabolites was performed as mentioned above before adding 200 μl of a uniformly [$^{13}$C]-labeled *E. coli* cell extract as internal quantification standard [70]. Then, metabolites were analyzed by IC-HRMS as described above with measured concentrations of metabolites expressed as a total cellular concentration assuming a volume of $10^8$ cells being equal to 5.8 μl [37].

## Immunofluorescence analyses

Cells were washed twice with PBS and fixed with 4% paraformaldehyde for 10 min at room temperature, spread on slides and permeabilised with 0.05% Triton X-100. After incubation in PBS containing 4% BSA overnight, cells were incubated for 45 min with anti-aldolase rabbit serum (Aldo; 1:1000). After washing with PBS, samples were incubated for 45 min with a secondary anti-rabbit IgG antibody conjugated to Alexa Fluor 594 (ThermoFisher). Slides were washed and mounted with SlowFade Gold (Molecular Probes). Images were acquired with the Metamorph software on a Zeiss Imager Z1 or an Axioplan 2 microscope as previously described [71].

## Fluorescence intensity ratio measurements

The microscopy experiments were performed at the Bordeaux Imaging Center (BIC) on an inverted Leica DMI 6000 microscope (Leica Microsystems, Wetzlar, Germany) equipped with a resolutive HQ2 camera (Photometrics, Tucson, USA). The illumination system used was a Lumencor spectra 7 (Lumencor, Beaverton, USA). The objective used was a HCX PL APO CS 63X oil 1.32 NA. This system was controlled by MetaMorph software (Molecular Devices, Sunnyvale, USA). Cells were both observed in differential interference contrast mode by transmission and fluorescence microscopy (CFP excitation/CFP emission for donor acquisition, CFP excitation/YFP emission for FRET acquisition). We quantified YFP (FRET) and CFP emission using the ImageJ software (NIH, USA).

## Fluorescence lifetime imaging microscopy (FLIM)

The FLIM measurements were performed at the BIC with the Lambert Instrument FLIM Attachment (LIFA, Lambert Instrument, Roden, the Netherlands) that allows the generation of lifetime images by using the frequency domain method. This system consists of a modulated intensified CCD camera $Li^2$ CAM MD, a modulated light excitation light source and a modulated GenIII image intensifier. For widefield epi-illumination, modulated LED (Light-Emitted-Diode) was used at 451 nm (3 W) for CFP excitation. Both the LED and the intensifier were modulated at frequency up to 100 MHz. A series of 12 images was recorded for each sample. By varying the phase shifts (12 times) between the illuminator and the intensifier modulation we calculated the phase and modulation for each pixel of the image. Consequently, we determined the sample fluorescence lifetime image using the manufacturer's software LI-FLIM software. Lifetimes were referenced to a solution of erythrosin B (1 mg ml$^{-1}$) set at 0.086 ns [72].

## Supporting information

**S1 Fig. Growth curves of other FBPase and/or PFK mutant cell lines.** This figure shows growth curves of the $^{RNAi}$PFK clone A8 and Δ*fbpase*/$^{RNAi}$PFK clone F8 mutant cell lines, tetracycline-induced or not, in glucose-rich (+Glc) and glucose-depleted (−Glc) conditions. Cells were maintained in the exponential growth phase (between $10^6$ and $10^7$ cells ml$^{-1}$), and cumulative cell numbers reflect normalization for dilution during cultivation. Similar growth curves were obtained for the $^{RNAi}$PFK clone A4 and Δ*fbpase*/$^{RNAi}$PFK clone A9 presented in Fig 3B. The data underlying this figure can be found in https//doi.org/10.5281/zenodo.15148560.
(TIF)

**S2 Fig. Validation of the Δ*sbpase* mutant cell line.** Panel **A** shows a PCR analysis of genomic DNA isolated from the parental (WT), Δ*sbpase::BSD*, Δ*sbpase::PAC* and Δ*sbpase::BSD*/Δ*sbpase::PAC* (named Δ*sbpase* in the manuscript) cell lines, using two primers flanking the *SBPase* gene (5′-ggaggtactttctcttctatttct-3′ and 5′-aagtcagagcacattacgaccac-3′). The 3 PCR products, SBPase, PAC, and BSD, correspond to the DNA fragments described in panel **B**. As expected, PCR amplification of the *SBPase* gene was not observed in Δ*sbpase::BSD*/Δ*sbpase::PAC* cell ine, *BSD* PCR-products were observed in the Δ*sbpase::BSD* and Δ*sbpase::BSD*/Δ*sbpase::PAC* cell lines, whereas *PAC* PCR-products were observed in the Δ*sbpase::PAC* and Δ*sbpase::BSD*/Δ*sbpase::PAC* cell lines. The data underlying this figure can be found in https//doi.org/10.5281/zenodo.15148560.
(TIF)

## Acknowledgments

We thank K. Ersfeld, (Hull University, UK) for the anti-Myc immune serum, H. Imamura (Yamaguchi University, Japan) for providing the ATeam sequence, and C. Pujol (Bordeaux Imaging Center—BIC, France) for her invaluable technical assistance for the FRET and FLIM experiments. We thank A. Zouine, V. Pitard, and J.-M. Griffon at the TBMCore facility

(FACSility, CNRS UAR 3427, INSERM US 05, Université de Bordeaux) for technical assistance, data acquisition on BD FACS Aria III Sorter, and interpretation. The microscopy was done in the Bordeaux Imaging Center (BIC), a service unit of the CNRS-INSERM and Bordeaux University, member of the national infrastructure France BioImaging.

## Author contributions

**Conceptualization:** Daniel Inaoka, Jean-Charles Portais, Frédéric Bringaud.

**Data curation:** Hanna Kulyk, Frédéric Bringaud.

**Formal analysis:** Nicolas Plazolles, Hanna Kulyk, Edern Cahoreau, Marc Biran.

**Funding acquisition:** Frédéric Bringaud.

**Methodology:** Nicolas Plazolles, Hanna Kulyk, Edern Cahoreau, Marc Biran, Marion Wargnies, Erika Pineda, Mohammad El Kadri, Aline Rimoldi, Perrine Hervé, Corinne Asencio.

**Supervision:** Loïc Rivière, Emmanuel Tetaud, Frédéric Bringaud.

**Writing – original draft:** Frédéric Bringaud.

**Writing – review & editing:** Loïc Rivière, Paul AM Michels, Daniel Inaoka, Emmanuel Tetaud, Jean-Charles Portais.

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
