## [Editor Report · Decision Letter 0]

12 Nov 2024

Dear Dr Bringaud, 

Thank you for submitting your manuscript entitled "The glycosomal ATP-dependent phosphofructokinase of Trypanosoma brucei operates in the gluconeogenic direction in cellulo under in vivo-like conditions" for consideration as a Research Article by PLOS Biology.

Your manuscript has now been evaluated by the PLOS Biology editorial staff, as well as by an academic editor with relevant expertise, and I am writing to let you know that we would like to send your submission out for external peer review.

Once your full submission is complete, your paper will undergo a series of checks in preparation for peer review. After your manuscript has passed the checks it will be sent out for review. To provide the metadata for your submission, please Login to Editorial Manager (https://www.editorialmanager.com/pbiology) within two working days, i.e. by Nov 14 2024 11:59PM.

Kind regards,

Melissa

Melissa Vazquez Hernandez, Ph.D.

Associate Editor

PLOS Biology

---

## [Decision Letter · Decision Letter 1]

20 Dec 2024

Dear Dr Bringaud,

Thank you for your patience while your manuscript "The glycosomal ATP-dependent phosphofructokinase of Trypanosoma brucei operates in the gluconeogenic direction in cellulo under in vivo-like conditions" went through peer-review at PLOS Biology. Your manuscript has now been evaluated by the PLOS Biology editors, an Academic Editor with relevant expertise, and by several independent reviewers.

As you will see in the reports, the reviewers are generally positive but have raised some concerns and provided suggestions for improvement. Reviewer #1 requests clarifications on experimental methods and the inclusion of missing statistical tests. Notably, it is unclear how many replicates were performed for each experiment. If the data represent repeats, please specify this in the text; however, if they are from a single replicate, additional experiments will be necessary to ensure robustness. Additionally, Reviewer #2 suggests incorporating dynamic models to evaluate various scenarios.

We expect to receive your revised manuscript within 2 months. Please email us (plosbiology@plos.org) if you have any questions or concerns, or would like to request an extension. 

**IMPORTANT - SUBMITTING YOUR REVISION**

*Resubmission Checklist*

*Published Peer Review*

*PLOS Data Policy*

*Blot and Gel Data Policy*

Sincerely,

Melissa

Melissa Vazquez Hernandez, Ph.D.

Associate Editor

PLOS Biology

REVIEWERS' COMMENTS:

Reviewer #1: 

The Authors found that, using glycerol as a carbon source, T. brucei procyclic forms produce fructose 6-phosphate from fructose 1,6-bisphosphate using fructose 1,6-bisphosphatase, sedoheptulose 1,7-bisphosphatase, and ATP-dependent phosphofructokinase (PFK). The function of PFK in gluconeogenesis rather than in glycolysis is novel and attributed to the supramolecular organization of the pathway within glycosomes that allows metabolic channeling and produces ATP to maintain the ATP/ADP glycosomal balance. This is a thoughtfully considered study. Overall, this manuscript could advance the field, but some clarifications are needed.

1. Results, (line 168). The paragraph starts with the description of previous results, but it is not clear whether it continues with the description of those results, or it starts describing new results ("This was determined…": when, before or in this work?). This section needs a better description of Fig. 2 to explain how the conclusions were reached considering what the figure is showing. It is not clear to me where it is shown in Fig. 2 that 91% of F1,6BP were fully [13C]-enriched from [U-13C] glycerol, compared to 48% from [U-13C] proline. The meaning of the different arrows in the figure should be indicated. What are the other bands that are not due to PFK? The western blot needs some indication of the MW of the bands and the full blot should be presented in a supplementary figure. The same needs to be done with all other western blot analyses in the paper (Figs. 3A, 4A, 5C, 6C, 7A and 9H). In addition, when quantifications of western blot results are presented, the Authors should provide information about the number of western blots (n), and the statistical treatment of the data. 

2. Why Fig. 3B does not show s.d.? Is it only one experiment? Fig. 3C shows error bars with no indication of what they are, the number of experiments (n) and the statistical test used to detect significant differences. Fig. 3D needs clarification regarding how the % enrichment is evident when seeing the figure. 

3. Fig. 4A. The first arrow does not indicate any band. Where is the genetic evidence of SBPase KO?

4. Does Fig. 5E show the results of a single experiment?

5. Fig. 8 needs to indicate meaning of error bars, number of experiments (n).

6. Fig. 9I and 9J need to indicate meaning of error bars, number of experiments (n), significance and statistical test used to detect significant differences.

7. Lines 729-730 refer to something underlined and in italics. What figures shows that?

8. Line 875. Are there video-microscopy experiments?

— — — 

Reviewer #2: 

The article of Plazolles et al. is a rich depiction of contributions of various enzymes to gluconeogenesis in the procyclic form of Trypanosoma brucei. It breaks the dogma that ATP-dependent phosphofructokinases cannot be used in GNG and simultaneously provides new information on contributions of sedoheptulose bisphosphates and fructose bisphosphatase. The work includes an extraordinarily rich diversity of approaches including IC-MS based metabolic profiling and NMR analysis of metabolic end products. An impressive array of multiple gene knockouts/knockdowns, exploitation of marker free CRISPR-cas approches and novel routes to ascertain concentrations of ATP/ADP in cellulo.

There are just a couple of things I would like to have a clearer view upon.

Firstly it is the physiological role of PFK in GNG in trypanosomes. Could a sentence be added clarifying why glycerol is so much better than proline as a substrate for study (other than its better incorporation in H6P. It would be useful to explicitly outline why that is so. 

Presumably proline would be the physiologically relevant substrate in tsetse fly (although might glycerol be more abundant than hitherto believed in tsetse, for instance via their fat bodies?).

Significant and effective models for glycolytic flux have been geenrated for bloodstream form trypanosomes. I appreciate that the current paper is not dependent upon modelling, but have the authors contemplated exploiting dynamic models of metabolism to ascertain a spectrum of scenarios around PFK working in either direction and consequences of titrating alternative starting substrated to probe any interplay between GNG vs glycolysis under different conditions?

Given the dogma-busting nature of the discovery of a PFK operating in GNG, some contemplation on whether this reflects the evolutionary history of the trypanosome's enzyme. If it evolved from a PPi dependent PFK is it possible it had an ancestral role in GNG, then its switch to use of ATP may have been possible due to the regulatory environment and possible concentration differentials within different sub-compartments (e.g. tunneling within an enzyme complex) that has permitted the change. What might expressing a PPi dependent PFK in trypanosomes do? (I'm not advocating for this experiment to be done here, just curious).

The paper is well presented, albeit with a number of imperfections in English language including (but not restricted to) the below

line 33: add comma after "through gluconeogenesis"

line 35: changed "showed" to "show"

Line 58: Remove "the" before "central carbon metabolism"

line 60: change to "tightly regulated allosterically"

line 79: remove "the" before "kidney"

Line 87: humans (with an "s")

line 102: "by analogy" better than "in analogy"

line 104: put comms around "for instance"

line 107: remove "the" before PCF

Fig 1 title, also remove "the" before PCF

line 139: add "the" before PPP

line 143: remove "the" before cofactor and add "the" before PPP

line 145: E4P (not EAP)

line 187: "was" not were fully enriched

line 305: remove "s" from differences

line 448: "in" not "into" glycosomes

line 544: remove "the" before procyclic

---

## [Editor Report · Decision Letter 2]

13 Mar 2025

Dear Frédéric,

Thank you for your patience while we considered your revised manuscript "The glycosomal ATP-dependent phosphofructokinase of Trypanosoma brucei operates in the gluconeogenic direction in cellulo under in vivo-like conditions" for publication as a Research Article at PLOS Biology. This revised version of your manuscript has been evaluated by the PLOS Biology editors, and the Academic Editor.

Based on our Academic Editor's assessment of your revision, we are likely to accept this manuscript for publication, provided you satisfactorily address the remaining editorial requests. Please make sure to address the following data and other policy-related requests.

a) We routinely suggest changes to titles to ensure maximum accessibility for a broad, non-specialist readership, and to ensure they reflect the contents of the paper. In this case, we would suggest a minor edit to the title, as follows. Please ensure you change both the manuscript file and the online submission system, as they need to match for final acceptance:

"The glycosomal ATP-dependent phosphofructokinase of Trypanosoma brucei operates in the gluconeogenic direction"

b) Please indicate what does the error bar indicate in Fig 8. 

Please supply the numerical values either in the a supplementary file or as a permanent DOI’d deposition for the following figures:

Figure 2, 3BCD, 4BC, 5DE, 7B, 8, 9CDIJ, S1

d) Please cite the location of the data clearly in all relevant main and supplementary Figure legends, e.g. “The data underlying this Figure can be found in S1 Data” or “The data underlying this Figure can be found in https://doi.org/10.5281/zenodo.XXXXX”

e) Please ensure that you are using best practice for statistical reporting and data presentation. These are our guidelines https://journals.plos.org/plosbiology/s/best-practices-in-research-reporting#loc-statistical-reporting and a useful resource on data presentation https://journals.plos.org/plosbiology/article?id=10.1371/journal.pbio.1002128

- If you are reporting experiments where n ≤ 5, please plot each individual data point.

f) We require the original, uncropped and minimally adjusted images supporting all blot and gel results reported in the Figures 1B, 2, 3A, 4A, 5BC, 6BC, 7A, 9H, S2A

We will require these files before a manuscript can be accepted so please prepare and upload them now. Please carefully read our guidelines for how to prepare and upload this data: https://journals.plos.org/plosbiology/s/figures#loc-blot-and-gel-reporting-requirements

g) Please ensure that your Data Statement in the submission system accurately describes where your data can be found and is in final format, as it will be published as written there.

h) Per journal policy, if you have generated any custom code during the course of this investigation, please make it available without restrictions upon publication. Please ensure that the code is sufficiently well documented and reusable, and that your Data Statement in the Editorial Manager submission system accurately describes where your code can be found.

We expect to receive your revised manuscript within two weeks. 

*Published Peer Review History*

*Press*

Sincerely,

Melissa

Melissa Vazquez Hernandez, Ph.D.

Associate Editor

PLOS Biology

---

## [Editor Report · Decision Letter 3]

10 Apr 2025

Dear Frédéric,

I would like to apologize for not answering your e-mail about the extension to submit the metadata, I somehow missed to get back to you. 

Thank you for the submission of your revised Research Article "The glycosomal ATP-dependent phosphofructokinase of Trypanosoma brucei also operates in the gluconeogenic direction" for publication in PLOS Biology. On behalf of my colleagues and the Academic Editor, Tania de Koning-Ward, I am pleased to say that we can in principle accept your manuscript for publication, provided you address any remaining formatting and reporting issues. These will be detailed in an email you should receive within 2-3 business days from our colleagues in the journal operations team; no action is required from you until then. Please note that we will not be able to formally accept your manuscript and schedule it for publication until you have completed any requested changes.

IMPORTANT: Many thanks for providing the images for all gels and blots. However, we require that these are the images of the full gels and not cropped or modified in any way. I am afraid that most of the images are still cropped. I have asked my colleagues to include this request alongside their own.

PRESS

Sincerely, 

Melissa

Melissa Vazquez Hernandez, Ph.D., Ph.D.

Associate Editor

PLOS Biology
